# INTERPRETABLE MODELS FOR GRANGER CAUSALITY USING SELF-EXPLAINING NEURAL NETWORKS

**Ričards Marcinkevičs**
Department of Computer Science
ETH Zürich
Universitätstrasse 6
8092 Zürich, Switzerland
ricards.marcinkevics@inf.ethz.ch

**Julia E. Vogt**
Department of Computer Science
ETH Zürich
Universitätstrasse 6
8092 Zürich, Switzerland
julia.vogt@inf.ethz.ch

## ABSTRACT

Exploratory analysis of time series data can yield a better understanding of complex dynamical systems. Granger causality is a practical framework for analysing interactions in sequential data, applied in a wide range of domains. In this paper, we propose a novel framework for inferring multivariate Granger causality under nonlinear dynamics based on an extension of self-explaining neural networks. This framework is more interpretable than other neural-network-based techniques for inferring Granger causality, since in addition to relational inference, it also allows detecting signs of Granger-causal effects and inspecting their variability over time. In comprehensive experiments on simulated data, we show that our framework performs on par with several powerful baseline methods at inferring Granger causality and that it achieves better performance at inferring interaction signs. The results suggest that our framework is a viable and more interpretable alternative to sparse-input neural networks for inferring Granger causality.

## 1 INTRODUCTION

Granger causality (GC) (Granger, 1969) is a popular practical approach for the analysis of multivariate time series and has become instrumental in exploratory analysis (McCracken, 2016) in various disciplines, such as neuroscience (Roebroeck et al., 2005), economics (Appiah, 2018), and climatology (Charakopoulos et al., 2018). Recently, the focus of the methodological research has been on inferring GC under nonlinear dynamics (Tank et al., 2018; Nauta et al., 2019; Wu et al., 2020; Khanna & Tan, 2020; Löwe et al., 2020), causal structures varying across replicates (Löwe et al., 2020), and unobserved confounding (Nauta et al., 2019; Löwe et al., 2020).

To the best of our knowledge, the latest powerful techniques for inferring GC do not target the effect sign detection (see Section 2.1 for a formal definition) or exploration of effect variability with time and, thus, have limited interpretability. This drawback defeats the purpose of GC analysis as an exploratory statistical tool. In some nonlinear interactions, one variable may have an exclusively positive or negative effect on another if it consistently drives the other variable up or down, respectively. Negative and positive causal relationships are common in many real-world systems, for example, gene regulatory networks feature inhibitory effects (Inoue et al., 2011) or in metabolomics, certain compounds may inhibit or promote synthesis of other metabolites (Rinschen et al., 2019). Differentiating between the two types of interactions would allow inferring and understanding such inhibition and promotion relationships in real-world dynamical systems and would facilitate a more comprehensive and insightful exploratory analysis. Therefore, we see a need for a framework capable of inferring nonlinear GC which is more amenable to interpretation than previously proposed methods (Tank et al., 2018; Nauta et al., 2019; Khanna & Tan, 2020). To this end, we introduce a novel method for detecting nonlinear multivariate Granger causality that is interpretable, in the sense that it allows detecting effect signs and exploring influences among variables throughout time. The main contributions of the paper are as follows:

1. We extend self-explaining neural network models (Alvarez-Melis & Jaakkola, 2018) to time series analysis. The resulting autoregressive model, named generalised vector autore-

gression (GVAR), is interpretable and allows exploring GC relations between variables, signs of Granger-causal effects, and their variability through time.

2. We propose a framework for inferring nonlinear multivariate GC that relies on a GVAR model with sparsity-inducing and time-smoothing penalties. Spurious associations are mitigated by finding relationships that are stable across original and time-reversed (Winkler et al., 2016) time series data.

3. We comprehensively compare the proposed framework and the powerful baseline methods of Tank et al. (2018), Nauta et al. (2019), and Khanna & Tan (2020) on a range of synthetic time series datasets with known Granger-causal relationships. We evaluate the ability of the methods to infer the ground truth GC structure and effect signs.

## 2 BACKGROUND AND RELATED WORK

### 2.1 GRANGER CAUSALITY

Granger-causal relationships are given by a set of directed dependencies within multivariate time series. The classical definition of Granger causality is given, for example, by Lütkepohl (2007). Below we define nonlinear multivariate GC, based on the adaptation by Tank et al. (2018). Consider a time series with $p$ variables: $\{\mathbf{x}_t\}_{t \in \mathbb{Z}^+} = \left\{ \left( \mathrm{x}_t^1 \ \mathrm{x}_t^2 \ ... \ \mathrm{x}_t^p \right)^\top \right\}_{t \in \mathbb{Z}^+}$. Assume that causal relationships between variables are given by the following structural equation model:

$$\mathrm{x}_t^i := g_i \left( \mathrm{x}_{1:(t-1)}^1, ..., \mathrm{x}_{1:(t-1)}^j, ..., \mathrm{x}_{1:(t-1)}^p \right) + \varepsilon_t^i, \text{ for } 1 \leq i \leq p, \tag{1}$$

where $\mathrm{x}_{1:(t-1)}^j$ is a shorthand notation for $\mathrm{x}_1^j, \mathrm{x}_2^j, ..., \mathrm{x}_{t-1}^j$; $\varepsilon_t^i$ are additive innovation terms; and $g_i(\cdot)$ are potentially nonlinear functions, specifying how the future values of variable $\mathrm{x}^i$ depend on the past values of $\mathbf{x}$. We then say that variable $\mathrm{x}^j$ does not Granger-cause variable $\mathrm{x}^i$, denoted as $\mathrm{x}^j \nrightarrow \mathrm{x}^i$, if and only if $g_i(\cdot)$ is constant in $\mathrm{x}_{1:(t-1)}^j$.

Depending on the form of the functional relationship $g_i(\cdot)$, we can also differentiate between positive and negative Granger-causal effects. In this paper, we define the effect sign as follows: if $g_i(\cdot)$ is increasing in all $\mathrm{x}_{1:(t-1)}^j$, then we say that variable $\mathrm{x}^j$ has a positive effect on $\mathrm{x}^i$, if $g_i(\cdot)$ is decreasing in $\mathrm{x}_{1:(t-1)}^j$, then $\mathrm{x}^j$ has a negative effect on $\mathrm{x}^i$. Note that an effect may be neither positive nor negative. For example, $\mathrm{x}^j$ can 'contribute' both positively and negatively to the future of $\mathrm{x}^i$ at different delays, or, for instance, the effect of $\mathrm{x}^j$ on $\mathrm{x}^i$ could depend on another variable.

Granger-causal relationships can be summarised by a directed graph $\mathcal{G} = (\mathcal{V}, \mathcal{E})$, referred to as summary graph (Peters et al., 2017), where $\mathcal{V} = \{1, ..., p\}$ is a set of vertices corresponding to variables, and $\mathcal{E} = \left\{ (i, j) : \mathrm{x}^i \rightarrow \mathrm{x}^j \right\}$ is a set of edges corresponding to Granger-causal relationships. Let $\boldsymbol{A} \in \{0, 1\}^{p \times p}$ denote the adjacency matrix of $\mathcal{G}$. The inference problem is then to estimate $\boldsymbol{A}$ from observations $\{\boldsymbol{x}_t\}_{t=1}^T$, where $T$ is the length of the time series observed. In practice, we usually fit a time series model that explicitly or implicitly infers dependencies between variables. Consequently, a statistical test for GC is performed. A conventional approach (Lütkepohl, 2007) used to test for linear Granger causality is the linear vector autoregression (VAR) (see Appendix A).

### 2.2 RELATED WORK

#### 2.2.1 TECHNIQUES FOR INFERRING NONLINEAR GRANGER CAUSALITY

Relational inference in time series has been studied extensively in statistics and machine learning. Early techniques for inferring undirected relationships include time-varying dynamic Bayesian networks (Song et al., 2009) and time-smoothed, regularised logistic regression with time-varying coefficients (Kolar et al., 2010). Recent approaches to inferring Granger-causal relationships leverage the expressive power of neural networks (Montalto et al., 2015; Wang et al., 2018; Tank et al., 2018; Nauta et al., 2019; Khanna & Tan, 2020; Wu et al., 2020; Löwe et al., 2020) and are often based on regularised autoregressive models, reminiscent of the Lasso Granger method (Arnold et al., 2007).

Tank et al. (2018) propose using sparse-input multilayer perceptron (cMLP) and long short-term memory (cLSTM) to model nonlinear autoregressive relationships within time series. Building on this, Khanna & Tan (2020) introduce a more sample efficient economy statistical recurrent unit (eSRU) architecture with sparse input layer weights. Nauta et al. (2019) propose a temporal causal discovery framework (TCDF) that leverages attention-based convolutional neural networks to test for GC. Appendix B contains further details about these and other relevant methods.

Approaches discussed above (Tank et al., 2018; Nauta et al., 2019; Khanna & Tan, 2020) and in Appendix B (Marinazzo et al., 2008; Ren et al., 2020; Montalto et al., 2015; Wang et al., 2018; Wu et al., 2020; Löwe et al., 2020) focus almost exclusively on relational inference and do not allow easily interpreting signs of GC effects and their variability through time. In this paper, we propose a more interpretable inference framework, building on self explaining-neural networks (Alvarez-Melis & Jaakkola, 2018), that, as shown by experiments, performs on par with the techniques described herein.

### 2.2.2 STABILITY-BASED SELECTION PROCEDURES

The literature on stability-based model selection is abundant (Ben-Hur et al., 2002; Lange et al., 2003; Meinshausen & Bühlmann, 2010; Sun et al., 2013). For example, Ben-Hur et al. (2002) propose measuring stability of clustering solutions under perturbations to assess structure in the data and select an appropriate number of clusters. Lange et al. (2003) propose a somewhat similar approach. Meinshausen & Bühlmann (2010) introduce the stability selection procedure applicable to a wide range of high-dimensional problems: their method guides the choice of the amount of regularisation based on the error rate control. Sun et al. (2013) investigate a similar procedure in the context of tuning penalised regression models.

### 2.2.3 SELF-EXPLAINING NEURAL NETWORKS

Alvarez-Melis & Jaakkola (2018) introduce self-explaining neural networks (SENN) – a class of intrinsically interpretable models motivated by explicitness, faithfulness, and stability properties. A SENN with a link function $g(\cdot)$ and interpretable basis concepts $\boldsymbol{h}(\boldsymbol{x}) : \mathbb{R}^p \to \mathbb{R}^k$ follows the form

$$f(\boldsymbol{x}) = g\left(\theta(\boldsymbol{x})_1 h(\boldsymbol{x})_1, ..., \theta(\boldsymbol{x})_k h(\boldsymbol{x})_k\right), \tag{2}$$

where $\boldsymbol{x} \in \mathbb{R}^p$ are predictors; and $\boldsymbol{\theta}(\cdot)$ is a neural network with $k$ outputs. We refer to $\boldsymbol{\theta}(\boldsymbol{x})$ as generalised coefficients for data point $\boldsymbol{x}$ and use them to 'explain' contributions of individual basis concepts to predictions. In the case of $g(\cdot)$ being sum and concepts being raw inputs, Equation 2 simplifies to

$$f(\boldsymbol{x}) = \sum_{j=1}^{p} \theta(\boldsymbol{x})_j x_j. \tag{3}$$

Appendix C lists additional properties SENNs need to satisfy, as defined by Alvarez-Melis & Jaakkola (2018).

A SENN is trained by minimising the following gradient-regularised loss function, which balances performance with interpretability:

$$\mathcal{L}_y(f(\boldsymbol{x}), y) + \lambda \mathcal{L}_{\boldsymbol{\theta}}\left(f(\boldsymbol{x})\right), \tag{4}$$

where $\mathcal{L}_y(f(\boldsymbol{x}), y)$ is a loss term for the ground classification or regression task; $\lambda > 0$ is a regularisation parameter; and $\mathcal{L}_{\boldsymbol{\theta}}(f(\boldsymbol{x})) = \left\|\nabla_{\boldsymbol{x}} f(\boldsymbol{x}) - \boldsymbol{\theta}(\boldsymbol{x})^{\top} \boldsymbol{J}_{\boldsymbol{x}}^{\boldsymbol{h}}(\boldsymbol{x})\right\|_2$ is the gradient penalty, where $\boldsymbol{J}_{\boldsymbol{x}}^{\boldsymbol{h}}$ is the Jacobian of $\boldsymbol{h}(\cdot)$ w.r.t. $\boldsymbol{x}$. This penalty encourages $f(\cdot)$ to be locally linear.

## 3 METHOD

We propose an extension of SENNs (Alvarez-Melis & Jaakkola, 2018) to autoregressive time series modelling, which is essentially a vector autoregression (see Equation 11 in Appendix A) with generalised coefficient matrices. We refer to this model as generalised vector autoregression (GVAR). The GVAR model of order $K$ is given by

$$\mathbf{x}_t = \sum_{k=1}^{K} \boldsymbol{\Psi}_{\boldsymbol{\theta}_k}\left(\mathbf{x}_{t-k}\right) \mathbf{x}_{t-k} + \boldsymbol{\varepsilon}_t, \tag{5}$$

where $\boldsymbol{\Psi}_{\boldsymbol{\theta}_k} : \mathbb{R}^p \to \mathbb{R}^{p \times p}$ is a neural network parameterised by $\boldsymbol{\theta}_k$. For brevity, we omit the intercept term here and in following equations. No specific distributional assumptions are made on the additive innovation terms $\boldsymbol{\varepsilon}_t$. $\boldsymbol{\Psi}_{\boldsymbol{\theta}_k}(\mathbf{x}_{t-k})$ is a matrix whose components correspond to the generalised coefficients for lag $k$ at time step $t$. In particular, the component $(i, j)$ of $\boldsymbol{\Psi}_{\boldsymbol{\theta}_k}(\mathbf{x}_{t-k})$ corresponds to the influence of $\mathbf{x}_{t-k}^j$ on $\mathbf{x}_t^i$. In our implementation, we use $K$ MLPs for $\boldsymbol{\Psi}_{\boldsymbol{\theta}_k}(\cdot)$ with $p$ input units and $p^2$ outputs each, which are then reshaped into an $\mathbb{R}^{p \times p}$ matrix. Observe that the model defined in Equation 5 takes on a form of SENN (see Equation 3) with future time series values as the response, past values as basis concepts, and sum as a link function.

Relationships between variables $\mathbf{x}^1, ..., \mathbf{x}^p$ and their variability throughout time can be explored by inspecting generalised coefficient matrices. To mitigate spurious inference in multivariate time series, we train GVAR by minimising the following penalised loss function with the mini-batch gradient descent:

$$\frac{1}{T-K} \sum_{t=K+1}^{T} \|\boldsymbol{x}_t - \hat{\boldsymbol{x}}_t\|_2^2 + \frac{\lambda}{T-K} \sum_{t=K+1}^{T} R(\boldsymbol{\Psi}_t) + \frac{\gamma}{T-K-1} \sum_{t=K+1}^{T-1} \|\boldsymbol{\Psi}_{t+1} - \boldsymbol{\Psi}_t\|_2^2, \quad (6)$$

where $\{\boldsymbol{x}_t\}_{t=1}^T$ is a single observed replicate of a $p$-variate time series of length $T$; $\hat{\boldsymbol{x}}_t = \sum_{k=1}^{K} \boldsymbol{\Psi}_{\hat{\boldsymbol{\theta}}_k}(\boldsymbol{x}_{t-k}) \boldsymbol{x}_{t-k}$ is the one-step forecast for the $t$-th time point by the GVAR model; $\boldsymbol{\Psi}_t$ is a shorthand notation for the concatenation of generalised coefficient matrices at the $t$-th time point: $\left[ \boldsymbol{\Psi}_{\hat{\boldsymbol{\theta}}_K}(\boldsymbol{x}_{t-K}) \ \boldsymbol{\Psi}_{\hat{\boldsymbol{\theta}}_{K-1}}(\boldsymbol{x}_{t-K+1}) \ ... \ \boldsymbol{\Psi}_{\hat{\boldsymbol{\theta}}_1}(\boldsymbol{x}_{t-1}) \right] \in \mathbb{R}^{p \times Kp}$; $R(\cdot)$ is a sparsity-inducing penalty term; and $\lambda, \gamma \geq 0$ are regularisation parameters. The loss function (see Equation 6) consists of three terms: *(i)* the mean squared error (MSE) loss, *(ii)* a sparsity-inducing regulariser, and *(iii)* the smoothing penalty term. Note, that in presence of categorically-valued variables the MSE term can be replaced with e.g. the cross-entropy loss.

The sparsity-inducing term $R(\cdot)$ is an appropriate penalty on the norm of the generalised coefficient matrices. Examples of possible penalties for the linear VAR are provided in Table 4 in Appendix A. These penalties can be easily adapted to the GVAR model. In the current implementation, we employ the elastic-net-style penalty term (Zou & Hastie, 2005; Nicholson et al., 2017) $R(\boldsymbol{\Psi}_t) = \alpha \|\boldsymbol{\Psi}_t\|_1 + (1 - \alpha) \|\boldsymbol{\Psi}_t\|_2^2$, with $\alpha = 0.5$.

The smoothing penalty term, given by $\frac{1}{T-K-1} \sum_{t=K+1}^{T-1} \|\boldsymbol{\Psi}_{t+1} - \boldsymbol{\Psi}_t\|_2^2$, is the average norm of the difference between generalised coefficient matrices for two consecutive time points. This penalty term encourages smoothness in the evolution of coefficients w.r.t. time and replaces the gradient penalty $\mathcal{L}_{\boldsymbol{\theta}}(f(\boldsymbol{x}))$ from the original formulation of SENN (see Equation 4). Observe that if the term is constrained to be 0, then the GVAR model behaves as a penalised linear VAR on the training data: coefficient matrices are invariant across time steps.

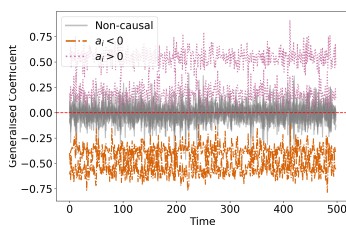

Figure 1: GVAR generalised coefficients inferred for a time series with linear dynamics.

Thus, the proposed penalised loss function (see Equation 6) allows controlling the *(i)* sparsity and *(ii)* nonlinearity of inferred autoregressive dependencies. As opposed to the related approaches of Tank et al. (2018) and Khanna & Tan (2020), signs of Granger causal effects and their variability in time can be assessed as well by interpreting matrices $\boldsymbol{\Psi}_{\hat{\boldsymbol{\theta}}_k}(\boldsymbol{x}_t)$, for $K + 1 \leq t \leq T$. Figure 1 shows a plot of generalised coefficients versus time in a toy linear time series (see Appendix L for details). Observe that for causal relationships, generalised coefficients are large in magnitude, whereas for non-causal links, coefficients are shrunk towards 0. Moreover, the signs of coefficients agree with the true interaction signs ($a_i$). We further support these claims with empirical results in Section 4. In addition, we provide an ablation study for the loss function in Appendix D.

## 3.1 INFERENCE FRAMEWORK

Once neural networks $\boldsymbol{\Psi}_{\hat{\boldsymbol{\theta}}_k}$, $k = 1, ..., K$, have been trained, we quantify strengths of Granger-causal relationships between variables by aggregating matrices $\boldsymbol{\Psi}_{\hat{\boldsymbol{\theta}}_k}(\boldsymbol{x}_t)$ across all time steps into

summary statistics. We aggregate the obtained generalised coefficients into matrix $\boldsymbol{S} \in \mathbb{R}^{p \times p}$ as follows:

$$S_{i,j} = \max_{1 \leq k \leq K} \left\{ \text{median}_{K+1 \leq t \leq T} \left( \left| \left( \boldsymbol{\Psi}_{\hat{\boldsymbol{\theta}}_k} \left( \boldsymbol{x}_t \right) \right)_{i,j} \right| \right) \right\}, \text{ for } 1 \leq i,j \leq p. \tag{7}$$

Intuitively, $S_{i,j}$ are statistics that quantify the strength of the Granger-causal effect of $\mathrm{x}^i$ on $\mathrm{x}^j$ using magnitudes of generalised coefficients. We expect $S_{i,j}$ to be close to 0 for non-causal relationships and $S_{i,j} \gg 0$ if $\mathrm{x}^i \rightarrow \mathrm{x}^j$. Note that in practice $\boldsymbol{S}$ is not binary-valued, as opposed to the ground truth adjacency matrix $\boldsymbol{A}$, which we want to infer, because the outputs of $\boldsymbol{\Psi}_{\hat{\boldsymbol{\theta}}_k}(\cdot)$ are not shrunk to exact zeros. Therefore, we need a procedure deciding for which variable pairs $S_{i,j}$ are significantly different from 0.

To infer a binary matrix of GC relationships, we propose a heuristic stability-based procedure that relies on time-reversed Granger causality (TRGC) (Haufe et al., 2012; Winkler et al., 2016). The intuition behind time reversal is to compare causality scores obtained from original and time-reversed data: we expect relationships to be flipped on time-reversed data (Haufe et al., 2012; Winkler et al., 2016). Winkler et al. (2016) prove the validity of time reversal for linear finite-order autoregressive processes. In our work, time reversal is leveraged for inferring stable dependency structures in nonlinear time series.

Algorithm 1 summarises the proposed stability-based thresholding procedure. During inference, two separate GVAR models are trained: one on the original time series data, and another on time-reversed data (lines 3-4 in Algorithm 1). Consequently, we estimate strengths of GC relationships with these two models, as in Equation 7, and choose a threshold for matrix $\boldsymbol{S}$ which yields the highest agreement between thresholded GC strengths estimated on original and time-reversed data (lines 5-9 in Algorithm 1). A sequence of $Q$ thresholds, given by $\boldsymbol{\xi} = (\xi_1, ..., \xi_Q)$, is considered where the $i$-th threshold is an $\xi_i$-quantile of values in $\boldsymbol{S}$. The agreement between inferred thresholded structures is measured (line 7 in Algorithm 1) using balanced accuracy score (Brodersen et al., 2010), denoted by $\text{BA}(\cdot, \cdot)$, equal-to the average of sensitivity and specificity, to reflect both sensitivity and specificity of the inference results. Other measures can be used for quantifying the agreement, for example, graph similarity scores (Zager & Verghese, 2008). In this paper, we utilise BA, because considered time series have sparse GC summary graphs and BA weighs positives and negatives equally. In practice, trivial solutions, such as inferring no causal relationships, only self-causal links or all possible causal links, are very stable. The agreement for such solutions is set to 0. Thus, the procedure assumes that the true causal structure is different from these trivial cases. Figure 6 in Appendix E contains an example of stability-based thresholding applied to simulated data.

---

**Algorithm 1:** Stability-based thresholding for inferring Granger causality with GVAR.

---

**Input:** One replicate of multivariate time series $\{\boldsymbol{x}_t\}_{t=1}^T$; regularisation parameters $\lambda$ and $\gamma \geq 0$; model order $K \geq 1$; sequence $\boldsymbol{\xi} = (\xi_1, ..., \xi_Q)$, $0 \leq \xi_1 < \xi_2 < ... < \xi_Q \leq 1$.

**Output:** Estimate $\hat{\boldsymbol{A}}$ of the adjacency matrix of the GC summary graph.

1 Let $\{\tilde{\boldsymbol{x}}_t\}_{t=1}^T$ be the time-reversed version of $\{\boldsymbol{x}_t\}_{t=1}^T$, i.e. $\{\tilde{\boldsymbol{x}}_1, ..., \tilde{\boldsymbol{x}}_T\} \equiv \{\boldsymbol{x}_T, ..., \boldsymbol{x}_1\}$.

2 Let $\tau(\boldsymbol{X}, \chi)$ be the elementwise thresholding operator. For each component of $\boldsymbol{X}$,
   $\tau(X_{i,j}, \chi) = 1$, if $|X_{i,j}| \geq \chi$, and $\tau(X_{i,j}, \chi) = 0$, otherwise.

3 Train an order $K$ GVAR with parameters $\lambda$ and $\gamma$ by minimising loss in Equation 6 on
   $\{\boldsymbol{x}_t\}_{t=1}^T$ and compute $\boldsymbol{S}$ as in Equation 7.

4 Train another GVAR on $\{\tilde{\boldsymbol{x}}_t\}_{t=1}^T$ and compute $\tilde{\boldsymbol{S}}$ as in Equation 7.

5 **for** $i = 1$ *to* $Q$ **do**

6 $\quad$ Let $\kappa_i = q_{\xi_i}(\boldsymbol{S})$ and $\tilde{\kappa}_i = q_{\xi_i}(\tilde{\boldsymbol{S}})$, where $q_\xi(X)$ denotes the $\xi$-quantile of $X$.

7 $\quad$ Evaluate agreement
   $$\varsigma_i = \tfrac{1}{2} \left[ \text{BA} \left( \tau(\boldsymbol{S}, \kappa_i), \tau\left(\tilde{\boldsymbol{S}}^\top, \tilde{\kappa}_i\right) \right) + \text{BA} \left( \tau\left(\tilde{\boldsymbol{S}}^\top, \tilde{\kappa}_i\right), \tau(\boldsymbol{S}, \kappa_i) \right) \right].$$

8 **end**

9 Let $i^* = \arg\max_{1 \leq i \leq Q} \varsigma_i$ and $\xi^* = \xi_{i^*}$.

10 Let $\hat{\boldsymbol{A}} = \tau(\boldsymbol{S}, q_{\xi^*}(\boldsymbol{S}))$.

11 **return** $\hat{\boldsymbol{A}}$.

---

To summarise, this procedure attempts to find a dependency structure that is stable across original and time-reversed data in order to identify significant Granger-causal relationships. In Section 4, we demonstrate the efficacy of this inference framework. In particular, we show that it performs on par with previously proposed approaches mentioned in Section 2.2.

### 3.1.1 COMPUTATIONAL COMPLEXITY

Our inference framework differs from the previously proposed cMLP, cLSTM (Tank et al., 2018), TCDF (Nauta et al., 2019), and eSRU (Khanna & Tan, 2020) w.r.t. computational complexity. Mentioned methods require training $p$ neural networks, one for each variable separately, whereas our inference framework trains $2K$ neural networks. A clear disadvantage of GVAR is its memory complexity: GVAR has many more parameters, since every MLP it trains has $p^2$ outputs. Appendix F provides a comparison between training times on simulated datasets with $p \in \{4, 15, 20\}$. In practice, for a moderate order $K$ and a larger $p$, we observe that training a GVAR model is faster than a cLSTM and eSRU.

## 4 EXPERIMENTS

The purpose of our experiments is twofold: (*i*) to compare methods in terms of their ability to infer the underlying GC structure; and (*ii*) to compare methods in terms of their ability to detect signs of GC effects. We compare GVAR to 5 baseline techniques: VAR with $F$-tests for Granger causality[1] and the Benjamini-Hochberg procedure (Benjamini & Hochberg, 1995) for controlling the false discovery rate (FDR) (at $q = 0.05$); cMLP and cLSTM (Tank et al., 2018)[2]; TCDF (Nauta et al., 2019)[3]; and eSRU (Khanna & Tan, 2020)[4]. We particularly focus on the baselines that, similarly to GVAR, leverage sparsity-inducing penalties, namely cMLP, cLSTM, and eSRU. In addition, we provide a comparison with dynamic Bayesian networks (Murphy & Russell, 2002) in Appendix I. The code is available in the GitHub repository: `https://github.com/i6092467/GVAR`.

### 4.1 INFERRING GRANGER CAUSALITY

We first compare methods w.r.t. their ability to infer GC relationships correctly on two synthetic datasets. We evaluate inferred dependencies on each independent replicate/simulation separately against the adjacency matrix of the ground truth GC graph, an example is shown in Figure 2. Each method is trained only on one sequence. Unless otherwise mentioned, we use accuracy (ACC) and balanced accuracy (BA) scores to evaluate thresholded inference results. For cMLP, cLSTM, and eSRU, the relevant weight norms are compared to 0. For TCDF, thresholding is performed within the framework based on the permutation test described by Nauta et al. (2019). For GVAR, thresholded matrices are obtained by applying Algorithm 1. In addition, we look at the continuously-valued inference results: norms of relevant weights, scores, and strengths of GC relationships (see Equation 7). We compare these scores against the true structure using areas under receiver operating characteristic (AUROC) and precision-recall (AUPRC) curves. For all evaluation metrics, we only consider off-diagonal elements of adjacency matrices, ignoring self-causal relationships, which are usually the easiest to infer. Note that our evaluation approach is different from those of Tank et al. (2018) and Khanna & Tan (2020); this partially explains some deviations from their re-

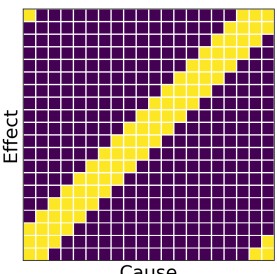

Figure 2: GC summary graph adjacency matrix of the Lorenz 96 system with $p = 20$. **Dark** cells correspond to the absence of a GC relationship; light cells denote a GC relationship.

sults. Relevant hyperparameters of all models are tuned to maximise the BA score or AUPRC (if a model fails to shrink any weights to zeros) by performing a grid search (see Appendix H for details about hyperparameter tuning). In Appendix M, we compare the prediction error of all models on held-out data.

---

[1] As implemented in the `statsmodels` library (Seabold & Perktold, 2010).
[2] `https://github.com/iancovert/Neural-GC`.
[3] `https://github.com/M-Nauta/TCDF`.
[4] `https://github.com/sakhanna/SRU_for_GCI`.

### 4.1.1 LORENZ 96 MODEL

A standard benchmark for the evaluation of GC inference techniques is the Lorenz 96 model (Lorenz, 1995). This continuous time dynamical system in $p$ variables is given by the following nonlinear differential equations:

$$\frac{d\mathbf{x}^i}{dt} = \left(\mathbf{x}^{i+1} - \mathbf{x}^{i-2}\right)\mathbf{x}^{i-1} - \mathbf{x}^i + F, \text{ for } 1 \leq i \leq p, \tag{8}$$

where $\mathbf{x}^0 := \mathbf{x}^p$, $\mathbf{x}^{-1} := \mathbf{x}^{p-1}$, and $\mathbf{x}^{p+1} := \mathbf{x}^1$; and $F$ is a forcing constant that, in combination with $p$, controls the nonlinearity of the system (Tank et al., 2018; Karimi & Paul, 2010). As can be seen from Equation 8, the true causal structure is quite sparse. Figure 2 shows the adjacency matrix of the summary graph for this dataset (for other datasets, adjacency matrices are visualised in Appendix G). We numerically simulate $R = 5$ replicates with $p = 20$ variables and $T = 500$ observations under $F = 10$ and $F = 40$. The setting is similar to the experiments of Tank et al. (2018) and Khanna & Tan (2020), but includes more variables.

Table 1 summarises the performance of the inference techniques on the Lorenz 96 time series under $F = 10$ and $F = 40$. For $F = 10$, all of the methods apart from TCDF are very successful at inferring GC relationships, even linear VAR. On average, GVAR outperforms all baselines, although performance differences are not considerable. For $F = 40$, the inference problem appears to be more difficult (Appendix J investigates performance of VAR and GVAR across a range of forcing constant values). In this case, TCDF and cLSTM perform surprisingly poorly, whereas cMLP, eSRU, and GVAR achieve somewhat comparable performance levels. GVAR attains the best combination of accuracy and BA scores, whereas cMLP has the highest AUROC and AUPRC. Thus, on Lorenz 96 data, the performance of GVAR is competitive with the other methods.

Table 1: Performance comparison on the Lorenz 96 model with $F = 10$ and $40$. Inference is performed on each replicate separately, standard deviations (SD) are evaluated across 5 replicates.

| $F$ | Model | ACC($\pm$SD) | BA($\pm$SD) | AUROC($\pm$SD) | AUPRC($\pm$SD) |
|---|---|---|---|---|---|
| 10 | VAR | 0.918($\pm$0.012) | 0.838($\pm$0.016) | 0.940($\pm$0.016) | 0.825($\pm$0.029) |
| | cMLP | 0.972($\pm$0.005) | 0.956($\pm$0.016) | 0.963($\pm$0.018) | 0.908($\pm$0.049) |
| | cLSTM | 0.970($\pm$0.010) | 0.950($\pm$0.028) | 0.958($\pm$0.029) | 0.925($\pm$0.050) |
| | TCDF | 0.871($\pm$0.012) | 0.709($\pm$0.044) | 0.857($\pm$0.027) | 0.601($\pm$0.053) |
| | eSRU | 0.966($\pm$0.011) | 0.951($\pm$0.021) | 0.963($\pm$0.020) | 0.936($\pm$0.034) |
| | GVAR (ours) | **0.982($\pm$0.003)** | **0.982($\pm$0.006)** | **0.997($\pm$0.001)** | **0.976($\pm$0.016)** |
| 40 | VAR | 0.864($\pm$0.008) | 0.585($\pm$0.028) | 0.745($\pm$0.047) | 0.474($\pm$0.036) |
| | cMLP | 0.683($\pm$0.027) | 0.805($\pm$0.017) | **0.979($\pm$0.016)** | **0.956($\pm$0.033)** |
| | cLSTM | 0.844($\pm$0.012) | 0.656($\pm$0.037) | 0.661($\pm$0.038) | 0.385($\pm$0.063) |
| | TCDF | 0.775($\pm$0.023) | 0.597($\pm$0.029) | 0.679($\pm$0.021) | 0.314($\pm$0.050) |
| | eSRU | 0.867($\pm$0.009) | **0.886($\pm$0.016)** | 0.934($\pm$0.021) | 0.834($\pm$0.033) |
| | GVAR (ours) | **0.945($\pm$0.010)** | 0.885($\pm$0.046) | 0.970($\pm$0.009) | 0.916($\pm$0.024) |

### 4.1.2 SIMULATED FMRI TIME SERIES

Another dataset we consider consists of rich and realistic simulations of blood-oxygen-level-dependent (BOLD) time series (Smith et al., 2011) that were generated using the dynamic causal modelling functional magnetic resonance imaging (fMRI) forward model. In these time series, variables represent 'activity' in different spatial regions of interest within the brain. Herein, we consider $R = 5$ replicates from the simulation no. 3 of the original dataset. These time series contain $p = 15$ variables and only $T = 200$ observations. The ground truth causal structure is very sparse (see Appendix G). Details about hyperparameter tuning performed for this dataset can be found in Appendix H.2. This experiment is similar to one presented by Khanna & Tan (2020).

Table 2 provides a comparison of the inference techniques. Surprisingly, TCDF outperforms other methods by a considerable margin (cf. Table 1). It is followed by our method that, on average, outperforms cMLP, cLSTM, and eSRU in terms of both AUROC and AUPRC. GVAR attains a BA score comparable to cLSTM. Importantly, eSRU fails to shrink any weights to exact zeros, thus, hindering the evaluation of accuracy and balanced accuracy scores (marked as 'NA' in Table 2). This

experiment demonstrates that the proximal gradient descent (Parikh & Boyd, 2014), as implemented by eSRU (Khanna & Tan, 2020), may fail to shrink any weights to 0 or shrinks all of them, even in relatively simple datasets. cMLP seems to provide little improvement over simple VAR w.r.t. AUROC or AUPRC. In general, this experiment promisingly shows that GVAR performs on par with the techniques proposed by Tank et al. (2018) and Khanna & Tan (2020) in a more realistic and data-scarce scenario than the Lorenz 96 experiment.

Table 2: Performance comparison on simulated fMRI time series. eSRU fails to shrink any weights to exact zeros, therefore, we have omitted accuracy and balanced accuracy score for it.

| Model | ACC($\pm$SD) | BA($\pm$SD) | AUROC($\pm$SD) | AUPRC($\pm$SD) |
|---|---|---|---|---|
| VAR | **0.910($\pm$0.006)** | 0.513($\pm$0.015) | 0.615($\pm$0.044) | 0.175($\pm$0.054) |
| cMLP | 0.846($\pm$0.025) | 0.614($\pm$0.068) | 0.616($\pm$0.068) | 0.191($\pm$0.058) |
| cLSTM | 0.830($\pm$0.022) | 0.655($\pm$0.053) | 0.663($\pm$0.051) | 0.234($\pm$0.058) |
| TCDF | 0.899($\pm$0.023) | **0.728($\pm$0.063)** | **0.812($\pm$0.041)** | **0.368($\pm$0.126)** |
| eSRU | NA | NA | 0.654($\pm$0.057) | 0.190($\pm$0.095) |
| GVAR (ours) | 0.806($\pm$0.070) | 0.652($\pm$0.045) | 0.687($\pm$0.066) | 0.289($\pm$0.116) |

## 4.2 INFERRING EFFECT SIGN

So far, we have only considered inferring GC relationships, but not the signs of Granger-causal effects. Such information can yield a better understanding of relations among variables. To this end, we consider the Lotka–Volterra model with multiple species $\big($Bacaër (2011) provides a definition of the original two-species system$\big)$, given by the following differential equations:

$$\frac{d\mathbf{x}^i}{dt} = \alpha \mathbf{x}^i - \beta \mathbf{x}^i \sum_{j \in Pa(\mathbf{x}^i)} \mathbf{y}^j - \eta \left(\mathbf{x}^i\right)^2, \text{ for } 1 \le i \le p, \tag{9}$$

$$\frac{d\mathbf{y}^j}{dt} = \delta \mathbf{y}^j \sum_{k \in Pa(\mathbf{y}^j)} \mathbf{x}^k - \rho \mathbf{y}^j, \text{ for } 1 \le j \le p, \tag{10}$$

where $\mathbf{x}^i$ correspond to population sizes of prey species; $\mathbf{y}^j$ denote population sizes of predator species; $\alpha, \beta, \eta, \delta, \rho > 0$ are fixed parameters controlling strengths of interactions; and $Pa(\mathbf{x}^i)$, $Pa(\mathbf{y}^j)$ are sets of Granger-causes of $\mathbf{x}^i$ and $\mathbf{y}^j$, respectively. According to Equations 9 and 10, the population size of each prey species $\mathbf{x}^i$ is driven down by $\left|Pa(\mathbf{x}^i)\right|$ predator species (negative effects), whereas each predator species $\mathbf{y}^j$ is driven up by $\left|Pa(\mathbf{y}^j)\right|$ prey populations (positive effects).

We simulate the multi-species Lotka–Volterra system numerically. Appendix K contains details about simulations and the summary graph of the time series. To infer effect directions, we inspect signs of median generalised coefficients for trained GVAR models. For cMLP, cLSTM, TCDF, and eSRU, we inspect signs of averaged weights in relevant layers. For VAR, we examine coefficient signs. For the sake of fair comparison, we restrict all models to a maximum lag of $K = 1$ (where applicable). In this experiment, we focus on BA scores for positive $\left(\text{BA}_{\text{pos}}\right)$ and negative $\left(\text{BA}_{\text{neg}}\right)$ relationships. Appendix L provides another example of detecting effect signs with GVAR, on a trivial benchmark with linear dynamics.

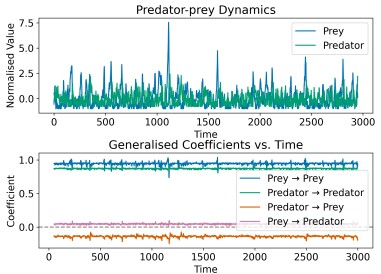

Figure 3: Simulated two-species Lotka–Volterra time series (top) and generalised coefficients (bottom). Prey have a positive effect on predators, and vice versa.

Table 3 shows the results for this experiment. Linear VAR does not perform well at inferring the GC structure, however, its coefficient signs are strongly associated with true signs of relationships. cMLP provides a considerable improvement in GC inference, and surprisingly its input weights are informative about the signs of GC effects. cLSTM fails to shrink any of the relevant weights to zero; furthermore, the signs of its weights are not associated with the true signs. Although eSRU performs better than VAR at inferring the summary graph, its weights are not associated with effect signs at

all. TCDF performs poorly in this experiment, failing to infer any relationships apart from self-causation. Our model considerably outperforms all baselines in detecting effect signs, achieving nearly perfect scores: it infers more meaningful and interpretable parameter values than all other models.

These results are not surprising, because the baseline methods, apart from linear VAR, rely on interpreting weights of relevant layers that, in general, do not need to be associated with effect signs and are only informative about the presence or absence of GC interactions. Since the GVAR model follows a form of SENNs (see Equation 2), its generalised coefficients shed more light into how the future of the target variable depends on the past of its predictors. This restricted structure is more intelligible and yet is sufficiently flexible to perform on par with sparse-input neural networks.

Table 3: Performance comparison on the multi-species Lotka–Volterra system. Next to accuracy and balanced accuracy scores, we evaluate BA scores for detecting positive and negative interactions.

| Model | ACC($\pm$SD) | BA($\pm$SD) | BA$_{pos}$($\pm$SD) | BA$_{neg}$($\pm$SD) |
|---|---|---|---|---|
| VAR | 0.383($\pm$0.095) | 0.635($\pm$0.060) | 0.845($\pm$0.024) | 0.781($\pm$0.042) |
| cMLP | 0.825($\pm$0.035) | 0.834($\pm$0.043) | 0.889($\pm$0.031) | 0.846($\pm$0.084) |
| cLSTM | NA | NA | 0.491($\pm$0.026) | 0.604($\pm$0.042) |
| TCDF | 0.832($\pm$0.013) | 0.500($\pm$0.012) | 0.538($\pm$0.045) | 0.504($\pm$0.090) |
| eSRU | 0.703($\pm$0.048) | 0.755($\pm$0.010) | 0.501($\pm$0.025) | 0.650($\pm$0.078) |
| GVAR (ours) | **0.977($\pm$0.005)** | **0.961($\pm$0.014)** | **0.932($\pm$0.027)** | **0.999($\pm$0.001)** |

In addition to inferring the summary graph, GVAR allows inspecting variability of generalised coefficients. Figure 3 provides an example of generalised coefficients inferred for a two-species Lotka–Volterra system. Although coefficients vary with time, GVAR consistently infers that the predator population is driven up by prey and the prey population is driven down by predators. For the multi-species system used to produce the quantitative results, inferred coefficients behave similarly (see Figure 12 in Appendix K).

## 5 CONCLUSION

In this paper, we focused on two problems: (*i*) inferring Granger-causal relationships in multivariate time series under nonlinear dynamics and (*ii*) inferring signs of Granger-causal relationships. We proposed a novel framework for GC inference based on autoregressive modelling with self-explaining neural networks and demonstrated that, on simulated data, its performance is promisingly competitive with the related methods of Tank et al. (2018) and Khanna & Tan (2020). Proximal gradient descent employed by cMLP, cLSTM, and eSRU often does not shrink weights to exact zeros and, thus, prevents treating the inference technique as a statistical hypothesis test. Our framework mitigates this problem by performing a stability-based selection of significant relationships, finding a GC structure that is stable on original and time-reversed data. Additionally, proposed GVAR model is more amenable to interpretation, since relationships between variables can be explored by inspecting generalised coefficients, which, as we showed empirically, are more informative than input layer weights. To conclude, the proposed model and inference framework are a viable alternative to previous techniques and are better suited for exploratory analysis of multivariate time series data.

In future research, we plan a thorough investigation of the stability-based thresholding procedure (see Algorithm 1) and of time-reversal for inferring GC. Furthermore, we would like to facilitate a more comprehensive comparison with the baselines on real-world data sets. It would also be interesting to consider better-informed link functions and basis concepts (see Equation 2). Last but not least, we plan to tackle the problem of inferring time-varying GC structures with the introduced framework.

### ACKNOWLEDGMENTS

We thank Djordje Miladinovic and Mark McMahon for valuable discussions and inputs. We also acknowledge Jonas Rothfuss and Kieran Chin-Cheong for their helpful feedback on the manuscript.

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

## A  Linear Vector Autoregression

Linear vector autoregression (VAR) (Lütkepohl, 2007) is a time series model conventionally used to test for Granger causality (see Section 2.1). VAR assumes that functions $g_i(\cdot)$ in Equation 1 are linear:

$$\mathbf{x}_t = \boldsymbol{\nu} + \sum_{k=1}^{K} \boldsymbol{\Psi}_k \mathbf{x}_{t-k} + \boldsymbol{\varepsilon}_t, \tag{11}$$

where $\boldsymbol{\nu} \in \mathbb{R}^p$ is the intercept vector; $\boldsymbol{\Psi}_k \in \mathbb{R}^{p \times p}$ are coefficient matrices; and $\boldsymbol{\varepsilon}_t \sim \mathcal{N}_p(\mathbf{0}, \boldsymbol{\Sigma_\varepsilon})$ are Gaussian innovation terms. Parameter $K$ is the order of the VAR model and determines the maximum lag at which Granger-causal interactions occur. In VAR, Granger causality is defined by zero constraints on the coefficients, in particular, $\mathbf{x}^i$ does not Granger-cause $\mathbf{x}^j$ if and only if, for all lags $k \in \{1, 2, ..., K\}$, $(\boldsymbol{\Psi}_k)_{j,i} = 0$. These constraints can be tested by performing, for example, $F$-test or Wald test.

Usually a VAR model is fitted using multivariate least squares. In high-dimensional time series, regularisation can be introduced to avoid inferring spurious associations. Table 4 shows various sparsity-inducing penalties for a linear VAR model of order $K$ (see Equation 11), described by Nicholson et al. (2017). Different penalties induce different sparsity patterns in coefficient matrices $\boldsymbol{\Psi}_1, \boldsymbol{\Psi}_2, ..., \boldsymbol{\Psi}_K$. These penalties can be adapted to the GVAR model for the sparsity-inducing term $R(\cdot)$ in Equation 6.

Table 4: Various sparsity-inducing penalty terms, described by Nicholson et al. (2017), for a linear VAR of order $K$. Herein, $\boldsymbol{\Psi} = \begin{bmatrix} \boldsymbol{\Psi}_1 & \boldsymbol{\Psi}_2 & ... & \boldsymbol{\Psi}_K \end{bmatrix} \in \mathbb{R}^{p \times Kp}$ (cf. Equation 11), and $\boldsymbol{\Psi}_{k:K} = \begin{bmatrix} \boldsymbol{\Psi}_k & \boldsymbol{\Psi}_{k+1} & ... & \boldsymbol{\Psi}_K \end{bmatrix}$. Different penalties induce different sparsity patterns in coefficient matrices.

| Model Structure | Penalty |
|---|---|
| Basic Lasso | $\lVert \boldsymbol{\Psi} \rVert_1$ |
| Elastic net | $\alpha \lVert \boldsymbol{\Psi} \rVert_1 + (1-\alpha) \lVert \boldsymbol{\Psi} \rVert_2^2, \alpha \in (0,1)$ |
| Lag group | $\sum_{k=1}^{K} \lVert \boldsymbol{\Psi}_k \rVert_F$ |
| Componentwise | $\sum_{i=1}^{p} \sum_{k=1}^{K} \lVert (\boldsymbol{\Psi}_{k:K})_i \rVert_2$ |
| Elementwise | $\sum_{i=1}^{p} \sum_{j=1}^{p} \sum_{k=1}^{K} \lVert (\boldsymbol{\Psi}_{k:K})_{i,j} \rVert_2$ |
| Lag-weighted Lasso | $\sum_{k=1}^{K} k^\alpha \lVert \boldsymbol{\Psi}_k \rVert_1, \alpha \in (0,1)$ |

## B  Inferring Granger Causality under Nonlinear Dynamics

Below we provide a more detailed overview of the related work on inferring nonlinear multivariate Granger causality, focusing on the recent machine learning techniques that tackle this problem.

**Kernel-based Methods**. Kernel-based GC inference techniques provide a natural extension of the VAR model, described in Appendix A, to nonlinear dynamics. Marinazzo et al. (2008) leverage reproducing kernel Hilbert spaces to infer linear Granger causality in an appropriate transformed feature space. Ren et al. (2020) introduce a kernel-based GC inference technique that relies on regularisation – Hilbert–Schmidt independence criterion (HSIC) Lasso GC.

**Neural Networks with Non-uniform Embedding**. Montalto et al. (2015) propose neural networks with non-uniform embedding (NUE). Significant Granger causes are identified using the NUE, a feature selection procedure. An MLP is 'grown' iteratively by greedily adding lagged predictor components as inputs. Once stopping conditions are satisfied, a predictor time series is claimed a significant cause of the target if at least one of its lagged components was added as an input. This technique is prohibitively costly, especially, in a high-dimensional setting, since it requires training and comparing many candidate models. Wang et al. (2018) extend the NUE by replacing MLPs with LSTMs.

**Neural Granger Causality**. Tank et al. (2018) propose inferring nonlinear Granger causality using structured multilayer perceptron and long short-term memory with sparse input layer weights, cMLP and cLSTM. To infer GC, $p$ models need to be trained with each variable as a response. cMLP and

cLSTM leverage the group Lasso penalty and proximal gradient descent (Parikh & Boyd, 2014) to infer GC relationships from trained input layer weights.

**Attention-based Convolutional Neural Networks**. Nauta et al. (2019) introduce the temporal causal discovery framework (TCDF) that utilises attention-based convolutional neural networks (CNN). Similarly to cMLP and cLSTM (Tank et al., 2018), the TCDF requires training $p$ neural network models to forecast each variable. Key distinctions of the TCDF are *(i)* the choice of the temporal convolutional network architecture over MLPs or LSTMs for time series forecasting and *(ii)* the use of the attention mechanism to perform attribution. In addition to the GC inference, the TCDF can detect time delays at which Granger-causal interactions occur. Furthermore, Nauta et al. (2019) provide a permutation-based procedure for evaluating variable importance and identifying significant causal links.

**Economy Statistical Recurrent Units**. Khanna & Tan (2020) propose an approach for inferring nonlinear Granger causality similar to cMLP and cLSTM (Tank et al., 2018). Likewise, they penalise norms of weights in some layers to induce sparsity. The key difference from the work of Tank et al. (2018) is the use of statistical recurrent units (SRUs) as a predictive model. Khanna & Tan (2020) propose a new sample-efficient architecture – economy-SRU (eSRU).

**Minimum Predictive Information Regularisation**. Wu et al. (2020) adopt an information-theoretic approach to Granger-causal discovery. They introduce learnable corruption, e.g. additive Gaussian noise with learnable variances, for predictor variables and minimise a loss function with minimum predictive information regularisation that encourages the corruption of predictor time series. Similarly to the approaches of Tank et al. (2018); Nauta et al. (2019); Khanna & Tan (2020), this framework requires training $p$ models separately.

**Amortised Causal Discovery & Neural Relational Inference**. Kipf et al. (2018) introduce the neural relational inference (NRI) model based on graph neural networks and variational autoencoders. The NRI model disentangles the dynamics and the undirected relational structure represented explicitly as a discrete latent graph variable. This allows pooling time series data with shared dynamics, but varying relational structures. Löwe et al. (2020) provide a natural extension of the NRI model to the Granger-causal discovery. They introduce a more general framework of the amortised causal discovery wherein time series replicates have a varying causal structure, but share dynamics. In contrast to the previous methods (Tank et al., 2018; Nauta et al., 2019; Khanna & Tan, 2020; Wu et al., 2020), which in this setting, have to be retrained separately for each replicate, the NRI is trained on the pooled dataset, leveraging shared dynamics.

## C  PROPERTIES OF SELF-EXPLAINING NEURAL NETWORKS

As defined by Alvarez-Melis & Jaakkola (2018), $g(\cdot)$, $\boldsymbol{\theta}(\cdot)$, and $\boldsymbol{h}(\cdot)$ in Equation 2 need to satisfy:

1. $g(\cdot)$ is monotonic and additively separable in its arguments;
2. $\frac{\partial g}{\partial z_i} > 0$ with $z_i = \theta(\boldsymbol{x})_i h(\boldsymbol{x})_i$, for all $i$;
3. $\boldsymbol{\theta}(\cdot)$ is locally difference-bounded by $\boldsymbol{h}(\cdot)$, i.e. for every $\boldsymbol{x}_0$, there exist $\delta > 0$ and $L \in \mathbb{R}$ s.t. if $\|\boldsymbol{x} - \boldsymbol{x}_0\| < \delta$, then $\|\boldsymbol{\theta}(\boldsymbol{x}) - \boldsymbol{\theta}(\boldsymbol{x}_0)\| \leq L \|\boldsymbol{h}(\boldsymbol{x}) - \boldsymbol{h}(\boldsymbol{x}_0)\|$;
4. $\{h(\boldsymbol{x})_i\}_{i=1}^{k}$ are interpretable representations of $\boldsymbol{x}$;
5. $k$ is small.

## D  ABLATION STUDY OF THE LOSS FUNCTION

We inspect hyperparameter tuning results for the GVAR model on Lorenz 96 (see Section 4.1.1) and synthetic fMRI time series (Smith et al., 2011) (see Section 4.1.2) as an ablation study for the loss function proposed (see Equation 6). Figures 4 and 5 show heat maps of BA scores (left) and AUPRCs (right) for different values of parameters $\lambda$ and $\gamma$ for Lorenz 96 and fMRI datasets, respectively. For the Lorenz 96 system, sparsity-inducing regularisation appears to be particularly important, nevertheless, there is also an increase in BA and AUPRC from a moderate smoothing penalty. For fMRI, we observe considerable performance gains from introducing both the sparsity-inducing and smoothing penalty terms. Given the sparsity of the ground truth GC structure and

the scarce number of observations ($T = 200$), these gains are not unexpected. During preliminary experiments, we ran grid search across wider ranges of $\lambda$ and $\gamma$ values, however, did not observe further improvements from stronger regularisation. In summary, these results empirically motivate the need for two different forms of regularisation leveraged by the GVAR loss function: the sparsity-inducing and smoothing penalty terms.

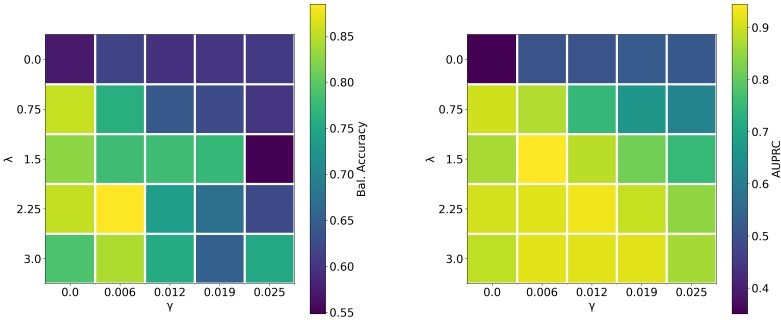

Figure 4: GVAR hyperparameter grid search results for Lorenz 96 time series (under $F = 40$) across 5 values of $\lambda \in [0.0, 3.0]$ and $\gamma \in [0.0, 0.02]$. Each cell shows average balanced accuracy (left) and AUPRC (right) across 5 replicates (**darker** colours correspond to lower performance) for one hyperparameter configuration.

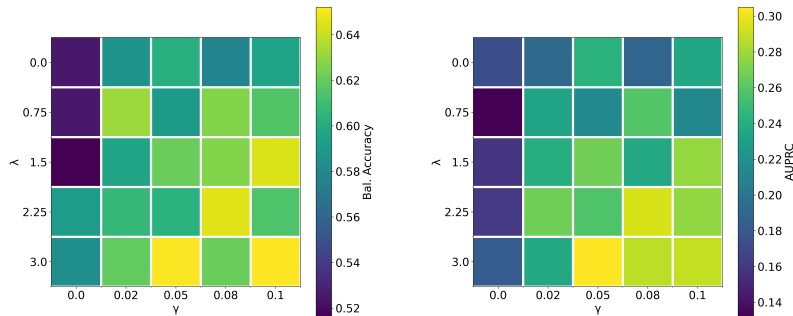

Figure 5: GVAR hyperparameter grid search results for simulated fMRI time series across 5 values of $\lambda \in [0.0, 3.0]$ and $\gamma \in [0.0, 0.1]$. The heat map on the left shows average BA scores, and the heat map on the right – average AUPRCs.

## E  STABILITY-BASED THRESHOLDING: EXAMPLE

Figure 6 shows an example of agreement between dependency structures inferred on original and time-reversed synthetic sequences across a range of thresholds (see Algorithm 1). In addition, we plot the BA score for resulting thresholded matrices evaluated against the true adjacency matrix. As can be seen, the peak of stability agrees with the highest BA achieved. In both cases, the procedure described by Algorithm 1 chooses the optimal threshold, which results in the highest agreement with the true dependency structure (unknown at the time of inference).

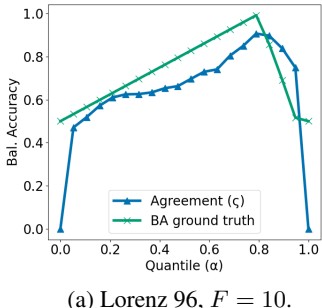 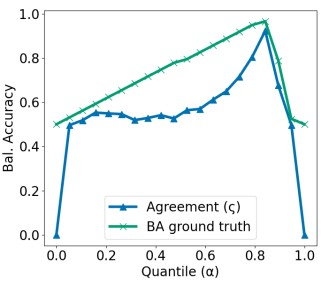

(a) Lorenz 96, $F = 10$.  (b) Multi-species Lotka–Volterra.

Figure 6: Agreement (▲) between GC structures inferred on the original and time-reversed data across a range of thresholds for one simulation of the Lorenz 96 (a) and multi-species Lotka–Volterra (b) systems. BA score (×) is evaluated against the ground truth adjacency matrix.

## F  COMPARISON OF TRAINING & INFERENCE TIME

To compare the considered methods in terms of their computational complexity, we measure training and inference time across three simulated datasets with $p \in \{4, 15, 20\}$ variables and varying time series lengths. This experiment was performed on an Intel Core i7-7500U CPU (2.70 GHz × 4) with a GeForce GTX 950M GPU. All models were trained for 1000 epochs with a mini-batch size of 64. In each dataset, the same numbers of hidden layers and hidden units were used across all models. When applicable, models were restricted to the same order ($K$). Table 5 contains average training and inference time in seconds with standard deviations. Observe that for the fMRI and Lorenz 96 datasets, GVAR is substantially faster than cLSTM and eSRU.

Table 5: Average training and inference time, in seconds, for the methods. Inference was performed on time series generated from the linear model (see Appendix L), simulated fMRI time series (see Section 4.1.2), and the Lorenz 96 system (see Section 4.1.1).

| Model | Linear ($p = 4, T = 500, K = 1$) | fMRI ($p = 15, T = 200, K = 1$) | Lorenz 96, $F = 10$ ($p = 20, T = 500, K = 5$) |
|---|---|---|---|
| VAR | 0.018(±0.001) | 0.27(±0.01) | 8.5(±0.6) |
| cMLP | 19.7(±3.5) | 55.1(±4.0) | 94.7(±2.4) |
| cLSTM | 999.2(±177.2) | 1763.3(±235.3) | 2023.8(±12.0) |
| TCDF | 11.4(±1.6) | 24.6(±1.7) | 50.1(±2.1) |
| eSRU | 62.9(±2.6) | 258.3(±31.1) | 671.4(±9.7) |
| GVAR (ours) | 75.5(±8.3) | 20.4(±0.7) | 197.7(±24.8) |

## G   GC SUMMARY GRAPHS OF SIMULATED TIME SERIES

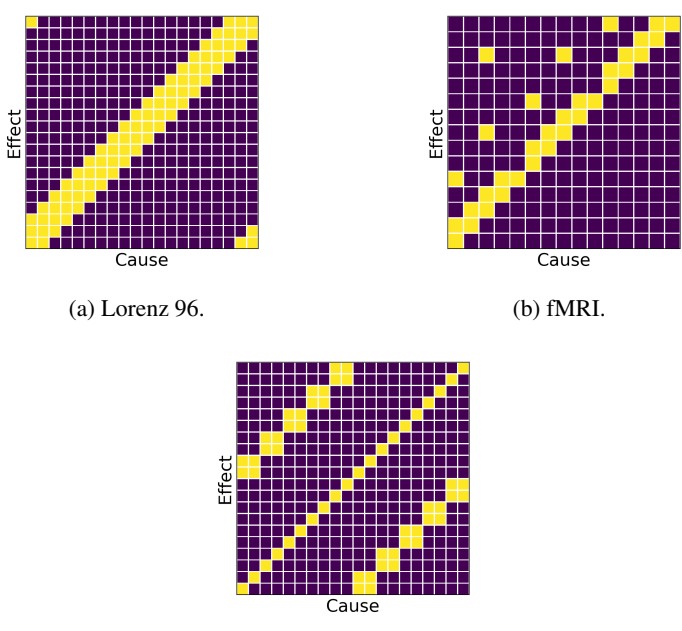

(a) Lorenz 96.                            (b) fMRI.

(c) Multi-species Lotka–Volterra.

Figure 7: Adjacency matrices of Granger-causal summary graphs for Lorenz 96 (see Section 4.1.1), simulated fMRI (see Section 4.1.2), and multi-species Lotka–Volterra (see Section 4.2) time series. **Dark** cells correspond to the absence of a GC relationship, i.e. $A_{i,j} = 0$; light cells denote a GC relationship, i.e. $A_{i,j} = 1$.

## H   HYPERPARAMETER TUNING

In our experiments (see Section 4), for all of the inference techniques compared, we searched across a grid of hyperparameters that control the sparsity of inferred GC structures. Other hyperparameters were fine-tuned manually. Final results reported in the paper correspond to the best hyperparameter configurations. With this testing setup, our goal was to fairly compare best achievable inferential performance of the techniques.

Tables 6, 7, and 8 provide ranges for hyperparameter values considered in each experiment. For cMLP and cLSTM (Tank et al., 2018), parameter $\lambda$ is the weight of the group Lasso penalty; for TCDF (Nauta et al., 2019), significance parameter $\alpha$ is used to decide which potential GC relationships are significant; eSRU (Khanna & Tan, 2020) has three different penalties weighted by $\lambda_{1:3}$. For the stability-based thresholding (see Algorithm 1) in GVAR, we used $Q = 20$ equally spaced values in $[0, 1]$ as sequence $\boldsymbol{\xi}$[5]. For Lorenz 96 and fMRI experiments, grid search results are plotted in Figures 4, 8, and 5. Figure 9 contains GVAR grid search results for the Lotka–Volterra experiment.

---

[5]We did not observe high sensitivity of performance w.r.t. $\boldsymbol{\xi}$, as long as sufficiently many evenly spaced sparsity levels are considered.

## H.1 LORENZ 96

Table 6: Hyperparameter values for Lorenz 96 datasets with $F = 10$ and $40$. Herein, $K$ denotes model order (maximum lag). If a hyperparameter is not applicable to a model, the corresponding entry is marked by 'NA'.

| Model | $K$ | # hidden layers | # hidden units | # training epochs | Learning rate | Mini-batch size | Sparsity hyperparam-s |
|---|---|---|---|---|---|---|---|
| VAR | 5 | NA | NA | NA | NA | NA | NA |
| cMLP | 5 | 2 | 50 | 1,000 | 1.0e-2 | NA | $F = 10$: $\lambda \in [0.5, 2.0]$; $F = 40$: $\lambda \in [0.0, 1.0]$ |
| cLSTM | NA | 2 | 50 | 1,000 | 5.0e-3 | NA | $F = 10$: $\lambda \in [0.1, 0.6]$; $F = 40$: $\lambda \in [0.2, 0.25]$ |
| TCDF | 5 | 2 | 50 | 1,000 | 1.0e-2 | 64 | $F = 10, 40$: $\alpha \in [0.0, 2.5]$ |
| eSRU | NA | 2 | 10 | 2,000 | 5.0e-3 | 64 | $F = 10, 40$: $\lambda_{1:3} \in [0.01, 0.1]$ |
| GVAR | 5 | 2 | 50 | 1,000 | 1.0e-4 | 64 | $F = 10, 40$: $\lambda \in [0.0, 3.0]$, $\gamma \in [0.0, 0.025]$ |

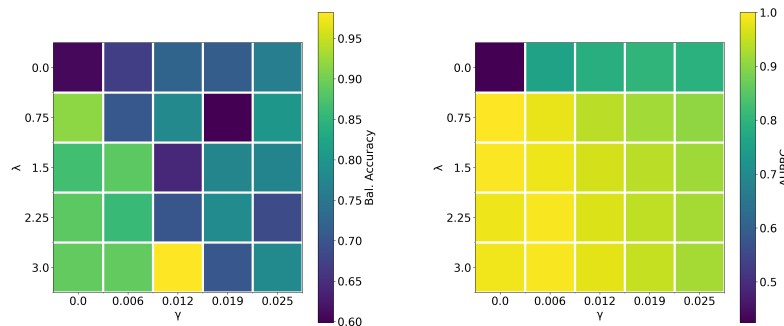

Figure 8: GVAR hyperparameter grid search results for Lorenz 96 time series, under $F = 10$, across 5 values of $\lambda \in [0.0, 3.0]$ and $\gamma \in [0.0, 0.02]$. Each cell shows average balanced accuracy (left) and AUPRC (right) across 5 replicates.

## H.2 FMRI

Table 7: Hyperparameter values for simulated fMRI time series.

| Model | $K$ | # hidden layers | # hidden units | # training epochs | Learning rate | Mini-batch size | Sparsity hyperparam-s |
|-------|-----|-----------------|----------------|-------------------|---------------|-----------------|------------------------|
| VAR | 1 | NA | NA | NA | NA | NA | NA |
| cMLP | 1 | 1 | 50 | 2,000 | 1.0e-2 | NA | $\lambda \in [0.001, 0.75]$ |
| cLSTM | NA | 1 | 50 | 1,000 | 1.0e-2 | NA | $\lambda \in [0.05, 0.3]$ |
| TCDF | 1 | 1 | 50 | 2,000 | 1.0e-3 | 64 | $\alpha \in [0.0, 2.0]$ |
| eSRU | NA | 2 | 10 | 2,000 | 1.0e-3 | 64 | $\lambda_1 \in [0.01, 0.05]$, $\lambda_2 \in [0.01, 0.05]$, $\lambda_3 \in [0.01, 1.0]$ |
| GVAR | 1 | 1 | 50 | 1,000 | 1.0e-4 | 64 | $\lambda \in [0.0, 3.0]$, $\gamma \in [0.0, 0.1]$ |

## H.3 LOTKA-VOLTERRA

Table 8: Hyperparameter values for multi-species Lotka–Volterra time series.

| Model | $K$ | # hidden layers | # hidden units | # training epochs | Learning rate | Mini-batch size | Sparsity hyperparam-s |
|-------|-----|-----------------|----------------|-------------------|---------------|-----------------|------------------------|
| VAR | 1 | NA | NA | NA | NA | NA | NA |
| cMLP | 1 | 2 | 50 | 2,000 | 5.0e-3 | NA | $\lambda \in [0.2, 0.4]$ |
| cLSTM | NA | 2 | 50 | 1,000 | 5.0e-3 | NA | $\lambda \in [0.0, 1.0]$ |
| TCDF | 1 | 2 | 50 | 2,000 | 1.0e-2 | 256 | $\alpha \in [0.0, 2.0]$ |
| eSRU | NA | 2 | 10 | 2,000 | 1.0e-3 | 256 | $\lambda_1 \in [0.01, 0.05]$, $\lambda_2 \in [0.01, 0.05]$, $\lambda_3 \in [0.01, 1.0]$ |
| GVAR | 1 | 2 | 50 | 500 | 1.0e-4 | 256 | $\lambda \in [0.0, 1.0]$, $\gamma \in [0.0, 0.01]$ |

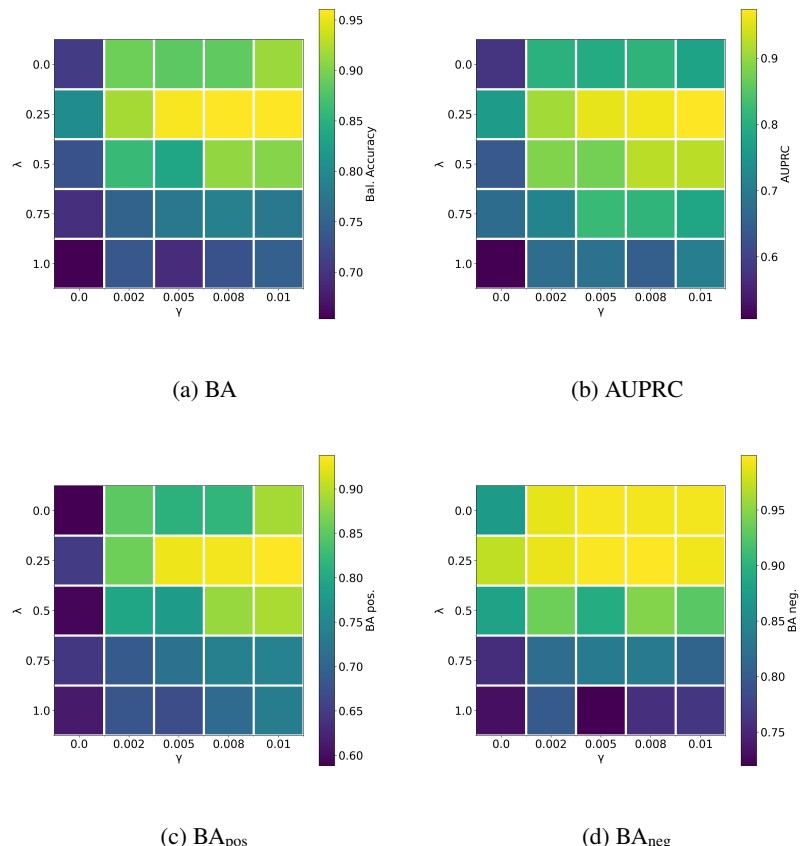

(a) BA

(b) AUPRC

(c) BA$_{pos}$

(d) BA$_{neg}$

Figure 9: GVAR hyperparameter grid search results for multi-species Lotka–Volterra time series across 5 values of $\lambda \in [0.0, 1.0]$ and $\gamma \in [0.0, 0.01]$. Heat maps above show balanced accuracies (a), AUPRCs (b), and balanced accuracies for positive (c) and negative (d) effects.

## I    COMPARISON WITH DYNAMIC BAYESIAN NETWORKS

We provide a comparison between GVAR and linear Gaussian dynamic Bayesian networks (DBN). DBNs are a classical approach to temporal structure learning (Murphy & Russell, 2002). We use R (R Core Team, 2020) package `dbnR` (Quesada, 2020) to fit DBNs on all datasets considered in Section 4. We use two structure learning algorithms: the max-min hill-climbing (MMHC) (Tsamardinos et al., 2006) and the particle swarm optimisation (Xing-Chen et al., 2007). Table 9 contains average balanced accuracies achieved by DBNs and GVAR for inferring the GC structure. Not surprisingly, DBNs outperform GVAR on the time series with linear dynamics, but fail to infer the true structure on Lorenz 96, fMRI, and Lotka–Volterra datasets.

Table 9: Comparison of balanced accuracy scores for GVAR and DBNs. Standard deviations (SD) are taken across 5 independent replicates.

| Model | Linear | Lorenz 96, $F = 10$ | Lorenz 96, $F = 40$ | fMRI | Lotka–Volterra |
|---|---|---|---|---|---|
| GVAR (ours) | 0.938($\pm$0.084) | **0.982($\pm$0.006)** | **0.885($\pm$0.046)** | **0.652($\pm$0.045)** | **0.961($\pm$0.014)** |
| DBN (MMHC) | 0.900($\pm$0.105) | 0.821($\pm$0.009) | 0.687($\pm$0.033) | 0.522($\pm$0.023) | 0.586($\pm$0.037) |
| DBN (PSO) | **0.950($\pm$0.028)** | 0.627($\pm$0.011) | 0.514($\pm$0.025) | 0.473($\pm$0.066) | 0.534($\pm$0.027) |

## J    THE LORENZ 96 SYSTEM: FURTHER EXPERIMENTS

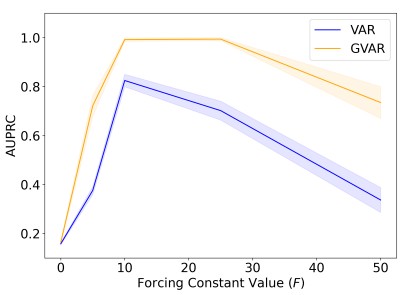

Figure 10: Inferential performance of GVAR across a range of forcing constant values.

In addition to the experiments in Section 4.1.1, we examine the performance of VAR and GVAR models across a range of forcing constant values $F = 0, 5, 10, 25, 50$ for the Lorenz 96 system with $p = 20$ variables. Figure 10 shows average AUPRCs with bands corresponding to the 95% CI for the mean. It appears that for both models, inference is more challenging for lower ($< 10$) and higher values of $F$ ($> 20$). This observation is in agreement with the results in Section 4.1.1, where all inference techniques performed worse under $F = 40$ than under $F = 10$. Note that herein same GVAR hyperparameters were used across all values of $F$. It is possible that better inferential performance could be achieved with GVAR after comprehensive hyperparameter tuning.

## K    THE LOTKA–VOLTERRA SYSTEM

The original Lotka–Volterra system (Bacaër, 2011) includes only one predator and one prey species, population sizes of which are denoted by x and y, respectively. Population dynamics are given by the following coupled differential equations:

$$\frac{d\mathrm{x}}{dt} = \alpha \mathrm{x} - \beta \mathrm{xy}, \tag{12}$$

$$\frac{d\mathrm{y}}{dt} = \delta \mathrm{yx} - \rho \mathrm{y}, \tag{13}$$

where $\alpha, \beta, \delta, \rho > 0$ are fixed parameters determining strengths of interactions.

In this paper, we consider a multiple species version of the system, given by Equations 9 and 10 in Section 4.2. We simulate the system under $\alpha = \rho = 1.1$, $\beta = \delta = 0.2$, $\eta = 2.75 \times 10^{-5}$, $\left|Pa(\mathrm{x}^i)\right| = \left|Pa(\mathrm{y}^j)\right| = 2$, $p = 10$, i.e. $2p = 20$ variables in total, with $T = 2000$ observations. Figure 11 depicts signs of GC effects between variables in a multi-species Lotka–Volterra with $2p = 20$ species and 2 parents per variable. We simulate this system numerically by using the Runge-Kutta method[6]. We make a few adjustments to the state transition equations, in particular: we introduce normally-distributed innovation terms to make simulated data noisy; during state transitions, we clip all population sizes below 0. Figure 12 shows traces of generalised coefficients inferred by GVAR: magnitudes and signs of coefficients reflect the true dependency structure.

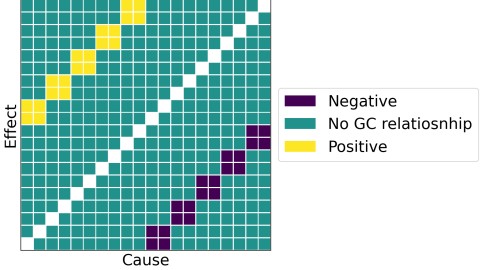

Figure 11: Signs of GC relationships between variables in the Lotka–Volterra system given by Equations 9 and 10, with $p = 10$. First ten columns correspond to prey species, whereas the last ten correspond to predators. Each prey species is 'hunted' by two predator species, and each predator species 'hunts' two prey species. Similarly to the other experiments, we ignore self-causal relationships.

---

[6]Simulations are based on the implementation available at `https://github.com/smkalami/lotka-volterra-in-python`.

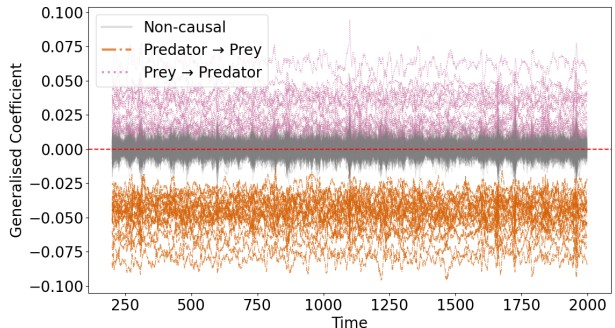

Figure 12: Variability of GVAR generalised coefficients throughout time for a simulation of the multi-species Lotka–Volterra system. Coefficients for **Granger non-causal** relationships fluctuate around 0; for Granger-causal relationships, coefficients are consistently different from 0: positive for **prey → predator** interactions and negative for **predator → prey**.

## L    EFFECT SIGN DETECTION IN A LINEAR VAR

Herein we provide results for the evaluation of GVAR and our inference framework on a very simple synthetic time series dataset. We simulate time series with $p = 4$ variables and linear interaction dynamics given by the following equations:

$$
\begin{aligned}
\mathrm{x}_t &= a_1 \mathrm{x}_{t-1} + \varepsilon_t^{\mathrm{x}}, \\
\mathrm{w}_t &= a_2 \mathrm{w}_{t-1} + a_3 \mathrm{x}_{t-1} + \varepsilon_t^{\mathrm{w}}, \\
\mathrm{y}_t &= a_4 \mathrm{y}_{t-1} + a_5 \mathrm{w}_{t-1} + \varepsilon_t^{y}, \\
\mathrm{z}_t &= a_6 \mathrm{z}_{t-1} + a_7 \mathrm{w}_{t-1} + a_8 \mathrm{y}_{t-1} + \varepsilon_t^{\mathrm{z}},
\end{aligned}
\tag{14}
$$

where coefficients $a_i \sim \mathcal{U}\left([-0.8, -0.2] \cup [0.2, 0.8]\right)$ are sampled independently in each simulation; and $\varepsilon_t^{\cdot} \sim \mathcal{N}\left(0, 0.16\right)$ are additive innovation terms. This is an adapted version of one of artificial datasets described by Peters et al. (2013), but without instantaneous effects.

The GC summary graph of the system is visualised in Figure 13. It is considerably denser than for the Lorenz 96, fMRI, and Lotka–Volterra time series investigated in Section 4.

Similarly to the experiment described in Section 4.2, we infer GC relationships with the proposed framework and evaluate inference results against the true dependency structure and effect signs. Table 10 contains average performance across 10 simulations achieved by GVAR with hyperparameter values $K = 1$, $\lambda = 0.2$, and $\gamma = 0.5$. In addition, we provide results for some of the baselines (no systematic hyperparameter tuning was performed for this experiment).

GVAR attains perfect AUROC and AUPRC in all 10 simulations. In some cases, stability-based thresholding fails to recover a completely correct GC structure, nevertheless, average accuracy and balanced accuracy scores are satisfactory. Signs of inferred generalised coefficients mostly agree with the ground truth effect signs, as given by coefficients $a_{1:8}$ in Equation 14.

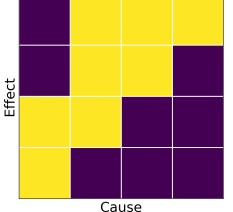

Figure 13: The adjacency matrix of the GC summary graph for the model given by Equation 14.

Not surprisingly, linear VAR performs the best on this dataset w.r.t. all evaluation metrics. Both cMLP and eSRU successfully infer GC relationships, achieving results comparable to GVAR. However, neither infers effect signs as well as GVAR. Thus, similarly to the experiment in Section 4.2, we conclude that generalised coefficients are more interpretable than neural network weights leveraged by cMLP, TCDF, and eSRU.

To summarise, this simple experiment serves as a sanity check and shows that our GC inference framework performs reasonably in low-dimensional time series with linear dynamics and a relatively dense GC summary graph (cf. Figure 7). Generally, the method successfully infers both the dependency structure and interaction signs.

Table 10: Performance on synthetic time series with linear dynamics, given by Equation 14. Averages and standard deviations are evaluated across 10 independent simulations. eSRU failed to shrink weights to exact 0s, therefore, we omit accuracy and BA scores for it.

| | VAR | cMLP | TCDF | eSRU | GVAR (ours) |
|---|---|---|---|---|---|
| **ACC** | **0.975($\pm$0.038)** | 0.867($\pm$0.085) | 0.791($\pm$0.056) | NA | 0.950($\pm$0.067) |
| **BA** | **0.981($\pm$0.029)** | 0.900($\pm$0.064) | 0.688($\pm$0.169) | NA | 0.938($\pm$0.084) |
| **AUROC** | **1.000($\pm$0.000)** | **1.000($\pm$0.000)** | 0.866($\pm$0.114) | 0.972($\pm$0.043) | **1.000($\pm$0.000)** |
| **AUPRC** | **1.000($\pm$0.000)** | **1.000($\pm$0.000)** | 0.812($\pm$0.128) | 0.967($\pm$0.046) | **1.000($\pm$0.000)** |
| **BA$_{pos}$** | **0.995($\pm$0.014)** | 0.761($\pm$0.206) | 0.574($\pm$0.235) | 0.613($\pm$0.168) | 0.920($\pm$0.158) |
| **BA$_{neg}$** | **0.990($\pm$0.019)** | 0.746($\pm$0.232) | 0.550($\pm$0.169) | 0.622($\pm$0.215) | 0.928($\pm$0.151) |

## M  PREDICTION ERROR

Herein we evaluate the prediction error of models on held-out data. Last 20% of time series points were held out to perform prediction on Lorenz 96, fMRI, and Lotka–Volterra datasets. Root-mean-square error (RMSE) was computed for predictions across $R = 5$ independent replicates:

$$\text{RMSE} = \frac{1}{p} \sum_{j=1}^{p} \sqrt{\frac{\sum_{t=1}^{T} \left( \hat{x}_t^j - x_t^j \right)}{T}}, \tag{15}$$

where $\hat{x}_t^j$ is the one-step forecast made by a model for the $t$-th point of the $j$-th variable, and $T$ is the length of the held-out time series segment. Table 11 contains average RMSEs for all models across the considered datasets. In general, RMSEs are not associated with the inferential performance of the models (cf. tables 1, 2, and 3). For example, while TCDF achieves the best inferential performance on fMRI (see Table 2), its prediction error is higher than for cMLP. This 'misalignment' between the prediction error and the consistency of variable selection is not surprising and has been discussed before, e.g. by Meinshausen & Bühlmann (2010).

Table 11: RMSEs of models on held-out data. Averages and standard deviations were taken across 5 independent replicates.

| Model | Lorenz 96, $F = 10$ | Lorenz 96, $F = 40$ | fMRI | Lotka–Volterra |
|---|---|---|---|---|
| VAR | 0.378($\pm$0.008) | 1.088($\pm$0.021) | 0.970($\pm$0.042) | 0.202($\pm$0.025) |
| cMLP | **0.336($\pm$0.030)** | **0.795($\pm$0.017)** | **0.724($\pm$0.037)** | **0.098($\pm$0.016)** |
| cLSTM | 0.592($\pm$0.013) | 0.983($\pm$0.014) | 0.874($\pm$0.045) | 0.691($\pm$0.035) |
| TCDF | 0.536($\pm$0.015) | 1.514($\pm$0.030) | 0.879($\pm$0.022) | 0.111($\pm$0.016) |
| eSRU | 1.000($\pm$0.010) | 1.006($\pm$0.015) | 1.000($\pm$0.048) | 0.720($\pm$0.035) |
| GVAR (ours) | 0.572($\pm$0.015) | 1.005($\pm$0.018) | 0.966($\pm$0.047) | 0.119($\pm$0.009) |

