# OpenReview forum: "Interpretable Models for Granger Causality Using Self-explaining Neural Networks"
_ICLR.cc/2021/Conference — ICLR 2021 Poster_

### Official Review · AnonReviewer3 · 2020-10-19
**The authors present GVAR which is a novel framework for inferring multivariate Granger causality under nonlinear dynamics based on an extension of SENNs..**

**Rating:** 6
**Confidence:** 4

**Review:**

# Introduction

The introduction is solid. My comment here is that there ought to be more discussion on sign detection. You leave even the definition until section 2.1, where a lay reader may struggle to understand what you are talking about if they are merely skimming your paper (as many do). Hence, for readability and for people to actually use this work, I suggest you focus more on sign detection early on and why it is important. You say that it is "important to differentiate between the two types of interactions" - well why is that?

# Background and related work

## 2.1

Stay clear of using $\mathbb{V}$ and $\mathbb{E}$ for vertices and edges. These type-settings are usually reserved for expectations and variances respectively.

# Method

Please give the form of SENN, in the same form as you've given your GVAR model, i.e. by way equation 4. They are difficult to compare otherwise. Right now what you claim to be an extension of SENN is not easily perceptible by reading section 2.2.3 and looking at equation 4.

The noise/innovation term $\boldsymbol{\epsilon}_t \in \mathcal{\mathbf{0},\mathbf{1}}$? As standard? Please specify.

At the bottom of this section you say "signs of Granger causal effects and their variability in time can be assessed as well by interpreting matrices" -- it would be helpful if you could provide a toy-example, of matrices, at this point to show us how precisely we are to interpret Psi to get to GC.

## 3.1

It is not clear _why_ you "aggregate the obtained generalised coefficients into matrix $\mathbf{S}"?

# Algorithm 1

What is $\boldsymbol \alpha$?

On line 7 you introduce the balance accuracy score. What is that? Why is it used in GVAR? Why did you choose that? What are the alternatives? My impression was that BA is the average of recall obtained on each class, in a classification problem. But here you are using it in a much more novel way. It would be very interesting to understand your reasoning and logic here.

Otherwise, well-written algorithm. Easy to follow and understand. My main pointer, as noted, is that you introduce far too many new things in the algorithm rather in the body itself. The algorithm is meant to summarise your method exposition of your method in the body, not introduce new items such as $\boldsymbol \alpha$ and BA.

# Experiments

What you are doing, from a purely ML point of view, is structure learning. This is very interesting of course, but I note that you have mainly compared to methods specifically focused on causality. It would be helpful/interesting if you also included more vanilla ML methods used for structure learning.

Spelling in sec 4.1 "releveant" should be 'relevant'.

If you can move some of your adjacency matrices into the main paper, from appendix F, that would help the paper a lot. Especially for the Lorenz-96 model. They visually demonstrate what you are trying to achieve, you barely need any legend to go with it. Right, for space issues presumably, they are wasted in the appendices. Please try to move at least one into the body as they a good opticals.

Why is the inference problem more difficult? F=8 is known to cause deterministic chaos but what about F=10 and 40? Would it not make more sense to evaluate a small ablation study on F from say 0 to 50 in increments of 10? It is not clear to me how computationally heavy GVAR is, hence this may be asking a lot. But since you have not added any such metrics, it is hard to know (computational complexity that is).

Your experiments shadow those of Khanna and Tan pretty well, yet you did not deploy GVAR on the DREAM-3IN SILICO NETWORK INFERENCE CHALLENGE? Why is that? That is the experiment where their model performs the worst, and since they do quite well (at least in Lorenz 96 as shown your paper) I would have thought that a good comparison too. Naturally, you only have so much space, but a comment would be helpful.

Please correct me if I am wrong here, but not one of your experiments contains an identfiable noise model, except for the model in appendix I? Why is the innovation N(0,0.16) that seems oddly specific? Why not standard white noise N(0,1)? And under this regime your models performs as well as VAR and CMLP by the looks of it.

---

> ### Author Response · Authors · 2020-11-16
> **Point-by-point Responses to Reviewer 3 (Part 1)**
>
> Thank you for the feedback. Below are our responses:
>
> *The introduction is solid. My comment here is that there ought to be more discussion on sign detection. You leave even the definition until section 2.1, where a lay reader may struggle to understand what you are talking about if they are merely skimming your paper (as many do). Hence, for readability and for people to actually use this work, I suggest you focus more on sign detection early on and why it is important. You say that it is "important to differentiate between the two types of interactions" - well why is that?*
> > We extended the introduction with an informal definition of positive and negative effects and stressed the importance of detecting these. Unfortunately, introducing effect signs formally at this stage would require formally defining GC, which, in our opinion, is too involved for the introduction.
> Regarding the importance of detecting positive and negative relationships, GC is a popular tool for exploratory analysis in various domains; it is applied to understand real world dynamical systems, e.g. connections between ROIs in the human brain. In such analyses, not only a presence or absence of a relationship is of interest, but also its form. Identifying positive and negative GC effects is one step towards the latter (arguably, more challenging) question.
>
> *Stay clear of using $\mathbb{V}$ and $\mathbb{E}$ for vertices and edges. These type-settings are usually reserved for expectations and variances respectively.*
> > In the revised version, we adjusted the notation to $\mathcal{V}$ and $\mathcal{E}$ to avoid potential confusion.
>
> *Please give the form of SENN, in the same form as you've given your GVAR model, i.e. by way equation 4. They are difficult to compare otherwise. Right now what you claim to be an extension of SENN is not easily perceptible by reading section 2.2.3 and looking at equation 4.*
> > We added a simplified equation for SENNs that is more comparable to the definition of the GVAR. Please see Eq. 3 in the revised version.
>
> *The noise/innovation term $\epsilon_t\in\boldsymbol{0},\boldsymbol{1}$? As standard? Please specify.*
> > Our method does not make specific distributional assumptions about the additive innovation term. We stated that clearly in the revised version.
>
> *At the bottom of this section you say "signs of Granger causal effects and their variability in time can be assessed as well by interpreting matrices" -- it would be helpful if you could provide a toy-example, of matrices, at this point to show us how precisely we are to interpret Psi to get to GC.*
> > We added a figure with a plot of generalised coefficients for a linear time series, see Figure 1. We hope this clarifies how coefficients can be interpreted to understand relationships.
>
> *It is not clear why you "aggregate the obtained generalised coefficients into matrix $\mathbf{S}$"?*
> > We added an explanation in the revised version, see Section 3.1. We aggregate coefficients into $\boldsymbol{S}$ to compute statistics, or causality scores, which could then be used to decide which relationships are significant. We would also like to note that other choices of statistics are possible and would be interesting to investigate in the future.
>
> *What is α?*
> > We renamed $\boldsymbol{\alpha}$ to $\boldsymbol{\xi}$, based on the feedback from other reviewers, in the revised version; henceforth, we will use $\boldsymbol{\xi}$ in our explanations. $\boldsymbol{\xi}$ is a sequence of values between $[0,1]$ that define thresholds considered by the Alg. 1. In particular, every threshold is a $\xi_i$-quantile of values in the matrix $\boldsymbol{S}$. We added an explanation of all of this in the revised version, see Section 3.1.
>
> *On line 7 you introduce the balance accuracy score. What is that? Why is it used in GVAR? Why did you choose that? What are the alternatives? My impression was that BA is the average of recall obtained on each class, in a classification problem. But here you are using it in a much more novel way. It would be very interesting to understand your reasoning and logic here.*
> > The balanced accuracy score (BA) is just an arithmetic mean between sensitivity and specificity. The score is conventionally used for model comparison in imbalanced classification problems. In our inference framework, the BA score is, indeed, used differently. We utilise it to compare two binarised matrices obtained from the original and time-reversed data. The intuition is that if sparsity level is correct, then the agreement between the two matrices should be high and the BA score will be high as well. Other measures are possible, for example, Manhattan distance, accuracy score, F1 score, graph similarity scores etc. We chose the BA score because it weighs 1s and 0s equally, and in many datasets we work with, the true structure is quite sparse (i.e. only few 1s). We added the explanation of our intuition to the revised manuscript as well, see Section 3.1.

---

> > ### Author Response · Authors · 2020-11-16
> > **Point-by-point Responses to Reviewer 3 (Part 2)**
> >
> > *Otherwise, well-written algorithm. Easy to follow and understand. My main pointer, as noted, is that you introduce far too many new things in the algorithm rather in the body itself. The algorithm is meant to summarise your method exposition of your method in the body, not introduce new items such as α and BA.*
> > > Please see the points above and the revised manuscript. We solidified the presentation of the algorithm to make it clearer and better connected with the pseudo-code.
> >
> > *What you are doing, from a purely ML point of view, is structure learning. This is very interesting of course, but I note that you have mainly compared to methods specifically focused on causality. It would be helpful/interesting if you also included more vanilla ML methods used for structure learning.*
> > > We agree that, ignoring the causality aspect, the underlying concept is that of structure learning. We added a comparison with dynamic Bayesian networks (DBN) in Appendix I of the revised manuscript. DBNs consider the time structure as well, and the comparison to them is meaningful in the context of detecting GC. To the best of our knowledge, the comparison to other structure learning algorithms would not be as fair in the context of this paper.
> >
> > *Spelling in sec 4.1 "releveant" should be 'relevant'.*
> > > We corrected this typo in the revised version.
> >
> > *If you can move some of your adjacency matrices into the main paper, from appendix F, that would help the paper a lot. Especially for the Lorenz-96 model. They visually demonstrate what you are trying to achieve, you barely need any legend to go with it. Right, for space issues presumably, they are wasted in the appendices. Please try to move at least one into the body as they a good opticals.*
> > > We moved a visualisation of one adjacency matrix to the main body of text in the revised version, see Figure 2.
> >
> > *Why is the inference problem more difficult? F=8 is known to cause deterministic chaos but what about F=10 and 40? Would it not make more sense to evaluate a small ablation study on F from say 0 to 50 in increments of 10? It is not clear to me how computationally heavy GVAR is, hence this may be asking a lot. But since you have not added any such metrics, it is hard to know (computational complexity that is).*
> > > We added a discussion of computational complexity to the main text as Section 3.1.1 and comparison between training and inference time of various models in Appendix F. These results illustrate how GVAR compares to other baselines in terms of computational complexity. In summary, GVAR is faster to train than cLSTM and eSRU. The complexity of our inference framework is quite different: GVAR trains $2K$ networks, whereas NN-based baselines train $p$ networks.
> > We performed an ablation experiment with the Lorenz 96 system, see Appendix J in the revised manuscript. We applied VAR and GVAR models to the Lorenz 96 under $F=0,5,10,25,50$. For both models, it appears that inference is more challenging under $F=0,5,$ and $50$ than under $F=10$. In the main experiments, $F=10$ and $40$ are considered to make the setting comparable to the experiments of Tank et al (2018) and Khanna & Tan (2020). An interesting comparison between $F=5$ and $F=10$ across a range of $p$ is provided by Karimi & Paul (2010). For $F=5$, they observe alternating windows of periodic and chaotic dynamics, whereas for $F=10$, they observe chaotic dynamics. Possibly, non-stationarity under $F=5$ (alternating windows of different behaviour) makes inference more challenging. Unfortunately, we did not find a principled comparison between higher and intermediate forcing (around $F=10$) in the literature. Inspecting time series visually, it appears that under $F=40$ time series are much ‘noisier’ than under $F=10$.

---

> > > ### Author Response · Authors · 2020-11-16
> > > **Point-by-point Responses to Reviewer 3 (Part 3)**
> > >
> > > *Your experiments shadow those of Khanna and Tan pretty well, yet you did not deploy GVAR on the DREAM-3IN SILICO NETWORK INFERENCE CHALLENGE? Why is that? That is the experiment where their model performs the worst, and since they do quite well (at least in Lorenz 96 as shown your paper) I would have thought that a good comparison too. Naturally, you only have so much space, but a comment would be helpful.*
> > > > We agree that DREAM3 would be an interesting experiment to conduct in the future. An important point of our current experiments, compared to Khanna & Tan (2020), was to additionally focus on effect sign detection. We think that effect sign detection is an important advantage of our model over eSRU and that both models perform, approximately, on par in terms of inference. Therefore, for the current paper, we did not think that the DREAM3 experiment would be particularly insightful. A bottleneck of our experiments was tuning the baseline models. Observe that Khanna & Tan (2020), similar to Tank et al (2018), focus their evaluation exclusively on AUROC and do not look at how the models shrink weights to exact 0s. In our experiments, we observed that even in moderately difficult datasets like fMRI it is very difficult to tune cLSTM and eSRU to converge to exact 0s. We expect that this will be even more challenging in DREAM3. Therefore, at the current stage, we do not think that DREAM3 would be an insightful experiment.
> > >
> > > *Please correct me if I am wrong here, but not one of your experiments contains an identifiable noise model, except for the model in appendix I? Why is the innovation N(0,0.16) that seems oddly specific? Why not standard white noise N(0,1)? And under this regime your model performs as well as VAR and CMLP by the looks of it.*
> > > > Indeed, we believe that the only experiment with an identifiable noise model is the linear VAR experiment from the Appendix. The noise term was adapted from an example by Peters et al (2013). We agree that the variance is oddly specific. Therefore, we rerun the experiment under $\mathcal{N}(0,1)$, but we did not observe any noteworthy differences. GVAR still achieves good results, comparable to VAR: average ACC of 0.950, average BA of 0.938, average AUROC of 1.000, average AUPRC of 1.000, average BA (pos.) of 0.920, and average BA (neg.) of 0.928.
> > > Under this regime GVAR performs as well as VAR and cMLP in terms of inferring the structure. However, observe that cMLP performs much worse at inferring effect signs than GVAR. This again illustrates that cMLP is not more interpretable than GVAR, even in this simple example.
> > >
> > > References:
> > >
> > > A. Tank, I. Covert, N. Foti, A. Shojaie, and E. Fox. Neural Granger causality for nonlinear time series, 2018. arXiv:1802.05842.
> > >
> > > S. Khanna and V. Y. F. Tan. Economy statistical recurrent units for inferring nonlinear Granger causality. In International Conference on Learning Representations, 2020.
> > >
> > > A. Karimi and M. R. Paul. Extensive chaos in the Lorenz-96 model. Chaos: An interdisciplinary journal of nonlinear science, 20(4):043105, 2010.
> > >
> > > J. Peters, D. Janzing, and B. Schölkopf.  Causal inference on time series using restricted structural equation models. In Advances in Neural Information Processing Systems 26, pp. 154–162. Curran Associates, Inc., 2013.

---

### Official Review · AnonReviewer1 · 2020-10-27
**Paper is  well presented but does not offer a fundamentally unique point of view as it combines existing ideas.**

**Rating:** 4
**Confidence:** 4

**Review:**

This paper primarily deals with learning Granger-causal relationships in multivariate time series in the nonlinear dynamics setting. The core method uses vector autoregressive modeling with sparsity inducing regularizers (elastic net and smoothness based fused lasso) along with the recently proposed with self-explaining neural networks (for interpretability). The authors also augment the framework by learning Granger-causal structures that are stable on original and time-reversed data. Exhaustive empirical analysis is done with recent GC baselines.  Some of my concerns with the paper are the following

a) Paper does seem a bit incremental in terms of using elastic-net and fused lasso based smoothness regularization within VAR framework. The idea of bringing SENN within this is useful but not enough to claim significant novelty.

b) Paper does not have scope of developing any useful theory ? Authors cite stability selection related work which has some decent guarantees on false discovery rate control. Can such theory be developed for the method proposed here ?

c) On the positive side the empirical analysis and coverage of related work is very exhaustive. Of particular significance is the result presented in Table 3. However, given the double computation and higher number of parameters involved in the method, I was expecting some discussion on time complexity and how to improve scalability of the method in practice.

---

> ### Author Response · Authors · 2020-11-16
> **Point-by-point Responses to Reviewer 1**
>
> Thank you for the constructive criticism. Below are our responses to your concerns:
>
> *a) Paper does seem a bit incremental in terms of using elastic-net and fused lasso based smoothness regularization within VAR framework. The idea of bringing SENN within this is useful but not enough to claim significant novelty.*
> > In response to this critique, we would like to stress some of the merits of our work:
> (i) The proposed inference framework is based on TRGC and stability. In the context of GC inference with NNs, none of these ideas appeared in the literature before. We observe that in practice the proximal gradient descent, as implemented in cMLP, cLSTM, and eSRU, often fails to shrink any weights to zeros (or shrinks all of them), whereas our procedure does not suffer from this problem, since variable selection is performed after training.
> (ii) To the best of our knowledge, none of the related work on neural-network-based inference of GC discusses the problem of inferring negative and positive effects. We believe the discussion of this problem would be useful to the community and the practitioners. GC has become an exploratory analysis tool in many domains. GVAR and the proposed inference framework can address questions often asked in such analyses, while inferring GC under nonlinear dynamics.
> For these reasons, we sincerely believe that our work adds fundamental value to the community.
>
> *b) Paper does not have scope of developing any useful theory? Authors cite stability selection related work which has some decent guarantees on false discovery rate control. Can such theory be developed for the method proposed here?*
> > For a more principled FDR control, an interesting path for this work would be to investigate the use of knockoff features, e.g. see Candes et al (2016), Jordon et al (2018). Deriving a way to generate knockoff features in multivariate time series and controlling for FDR with the help of the generated knockoffs would be an interesting further step for a more theoretically principled approach in the scope of this work.
>
> *c) On the positive side the empirical analysis and coverage of related work is very exhaustive. Of particular significance is the result presented in Table 3. However, given the double computation and higher number of parameters involved in the method, I was expecting some discussion on time complexity and how to improve scalability of the method in practice.*
> > We amended a discussion of the computational complexity to the revised manuscript in Section 3.1.1. The key difference is that all NN-based baselines require training $p$ neural networks (one for predicting each variable), whereas GVAR requires training $2K$ networks. Thus, under moderate order ($K$) and larger $p$, GVAR can even be faster to train than the baselines. In empirical experiments that we added in Appendix F, we observe that GVAR is often substantially faster than eSRU or cLSTM. We also believe that scalability can be improved by constructing good basis concepts across lagged values of each time series and then training only two neural networks on these concepts (on original and time-reversed data).
>
> References:
>
> Candes, E., Fan, Y., Janson, L., & Lv, J. (2016). Panning for gold: Model-X knockoffs for high-dimensional controlled variable selection. arXiv preprint arXiv:1610.02351.
>
> Jordon, J., Yoon, J., & van der Schaar, M. (2018). KnockoffGAN: Generating knockoffs for feature selection using generative adversarial networks. In International Conference on Learning Representations.

---

### Official Review · AnonReviewer4 · 2020-10-28
**Successful application of SENNs  to nonlinear dynamical systems infers GC relationships including sign in multi-variate time series**

**Rating:** 8
**Confidence:** 4

**Review:**

Interpretable ML approaches are very important to advance a wide variety of fields in healthcare and science. This work makes a significant contribution in this direction by successfully applying SENNs to multi-variate non-linear time series data. The method which the authors named GVAR is generic enough and can be applied to many different problems involving multi-variate time series data thereby making the paper quite impactful and significant. The authors look into GC relationships between variables including the sign of the relationship which they can successfully infer using their approach. The authors provide an array of favorable results across multiple simulated non-linear dynamical systems which corroborate their findings convincingly and also compare their results to a comprehensive set of related and SOTA approaches. There are no obvious inconsistencies or errors that I can see in the paper to the best of my knowledge. The paper is well written and conveys the information clearly enough. An exciting next step is to apply this approach to real data, could be a great addition to this paper but not necessary for publication.

---

> ### Author Response · Authors · 2020-11-16
> **Response to Reviewer 4**
>
> Thank you for seeing merits in our work! We agree that interpretability is crucial for advancing ML in various application domains. Indeed, an exciting next step would be to apply GVAR to real-world data. A particularly interesting application would be to study patient trajectories and relationships between observed variables within them, to understand and explore progressions of diseases and drivers of adverse of events.

---

### Official Review · AnonReviewer2 · 2020-10-28
**Marginally above acceptance threshold**

**Rating:** 6
**Confidence:** 4

**Review:**

Summary:
The paper proposes an interpretable method to detect Granger causality (GC) under nonlinear dynamics which can detect signs of Granger-causal effects (positive and negative) and inspect their variability over time. The novelty of this paper is 1 and 2 in the Pros below, and the methods utilized the existing stable frameworks such as a heuristic stability-based procedure that relies on time-reversed Granger causality (TRGC) (Winkler et al., 2016).

Reasons for score:
Although the motivation and the experiments were clear and the code was reproducible, the presentations such as in the results were sometimes unclear (below).
I think this would be a valuable paper in this community, the presentation and investigation may be required to obtain a higher rating.

Pros:
1. The method called generalized vector autoregression (GVAR) generalizes the VAR which is the basis of linear GC methods. The method is interpretable based on a self-explaining neural network (SENN) framework and allows exploring signs of Granger-causal effects and their variability through time.
2. The method utilizes a framework for inferring nonlinear multivariate GC that relies on a GVAR model with sparsity-inducing and time-smoothing penalties.
3. The method outperformed the baseline methods on a range of synthetic Lorenz 96 and Lotka–Volterra datasets in inferring the ground truth GC structure and effect signs.

Cons:
1. The detailed explanation and the difference from the previous TRGC work were unclear (below).
2. There was little discussion about the reason why TCDF outperformed the proposed methods on the simulated fMRI dataset.
3. There were no results in cost function (e.g., prediction error) even in the appendix. This detecting Granger causality would be unsupervised learning, thus the learning results based on the cost function are unknown. The information might help us understand the reason for 2.

Other comments:

Introduction:
“varying causal structures (Lowe et al., 2020)”: Did this paper focus on the varying causal structures?

Method:
The alpha in Alg.1 (threshold) is confusing with an elastic net regularization parameter.

There is no explanation of Q of Alg. 1 in the main text. Is this a variable threshold in the analogous way of the ROC curve? How was the Q determined?
Is the Alg. 1 the proposed method? The title “Stability-based thresholding” may be one of the components of the proposed method.

Experiments:

The number of sequences to evaluate the methods in each dataset was unknown. In my understanding, the proposed methods learn each model for each sequence like the methods of Tank et al. (2018) and Khanna & Tan (2020). In the shared code, for example, Lorenz 96 used 5 sequences (in the code, called datasets). This may be critical information to evaluate the method.

---

> ### Author Response · Authors · 2020-11-16
> **Point-by-point Responses to Reviewer 2 (Part 1)**
>
> We thank you for your feedback. Below are point-by-point responses to your concerns:
>
> *The detailed explanation and the difference from the previous TRGC work were unclear (below).*
> > We revised the manuscript and stressed the differences from the work of Winkler et al (2016) on TRGC, see Section 3.1 in the revised version. To the best of our knowledge, time reversal for inferring GC was considered for the first time by Haufe et al (2012). The intuition is to stress causality scores obtained on the original and time-reversed data, to make inference more robust. Intuitively, we expect that on time-reversed data inferred causal relationships will be flipped. Winkler et al (2016) focus exclusively on linear autoregressive processes and prove the validity of TRGC in this scope. Our work, on the other hand, focuses on nonlinear dynamics and leverages the time reversal trick described by Haufe et al (2012) and Winkler et al (2016) as a step within the proposed stability-based thresholding procedure, to identify which relationships are significant.
>
> *There was little discussion about the reason why TCDF outperformed the proposed methods on the simulated fMRI dataset.*
> > We were also surprised about the good performance of the TCDF on fMRI data, given its comparatively poor performance in other experiments. A possible reason  could be that diluted causal convolutions utilised by the TCDF are a useful inductive bias in this dataset. We also investigated prediction errors of all models on held-out data, but they do not explain a better performance of the TCDF (see below).
>
> *There were no results in cost function (e.g., prediction error) even in the appendix. This detecting Granger causality would be unsupervised learning, thus the learning results based on the cost function are unknown. The information might help us understand the reason for 2.*
> > We investigated the prediction errors of all models on held-out data. The discussion is provided in Appendix M of the revised manuscript. In general, we do not think that the prediction error is a good proxy for inferential performance, i.e. models that are good at predicting on held-out data are not necessarily good at inference. The misalignment between variable selection and minimisation of prediction error was discussed, for example, by Meinsausen and Bühlmann (2010).
> In our experiments, we observe that low prediction error is not associated with inferring the true GC structure well (see Appendix M). For example, while TCDF achieves the best inferential performance on fMRI, it does not have the lowest prediction error among the considered models. Thus, investigating prediction error, unfortunately, did not prove useful to understand why TCDF is better than other models at inferring the structure of fMRI time series.
>
> *Introduction: “varying causal structures (Lowe et al., 2020)”: Did this paper focus on the varying causal structures?*
> > Löwe et al (2020) investigate learning from multiple sequences wherein causal structure varies across sequences, but not across time. We adjusted the sentence to reflect this nuance in the revised version of the manuscript (see Section 1).
>
> *Method: The alpha in Alg.1 (threshold) is confusing with an elastic net regularization parameter.*
> > We adjusted the notation and replaced $\boldsymbol{\alpha}$ with $\boldsymbol{\xi}$, to avoid potential confusion with the elastic net hyperparameter.
>
> *There is no explanation of Q of Alg. 1 in the main text. Is this a variable threshold in the analogous way of the ROC curve? How was the Q determined? Is the Alg. 1 the proposed method? The title “Stability-based thresholding” may be one of the components of the proposed method.*
> > We added an explanation of $Q$ in Alg. 1 to the main text of the revised manuscript, see Section 3.1, and, in general, solidified the presentation of the Alg. 1. $Q$ is the number of thresholds considered. The algorithm considers $Q$ thresholds given by $\xi_i$-quantiles of the values in the matrix $\boldsymbol{S}$, for $i$ from 1 through $Q$. Indeed, this part of the algorithm is analogous to how the ROC curve is constructed. In Appendix H of the revised manuscript, we mention that we chose $Q=20$ in all of our experiments, and $\boldsymbol{\xi}$ was chosen to be an equispaced sequence of $Q$ values in $[0, 1]$. During our experiments, we did not observe sensitivity of inferential performance w.r.t. to the choice of $Q$, as long as $Q$ was sufficiently large, allowing to consider a fine-grained sequence of thresholds.
> Alg. 1, named “stability-based thresholding”, is a part of our inference framework that is used to identify significant relationships after training models on original and time-reversed data.

---

> > ### Author Response · Authors · 2020-11-16
> > **Point-by-point Responses to Reviewer 2 (Part 2)**
> >
> > *The number of sequences to evaluate the methods in each dataset was unknown. In my understanding, the proposed methods learn each model for each sequence like the methods of Tank et al. (2018) and Khanna & Tan (2020). In the shared code, for example, Lorenz 96 used 5 sequences (in the code, called datasets). This may be critical information to evaluate the method.*
> > > In sections 4.1.1 and 4.1.2, we mention that $R=5$ replicates, i.e. sequences, are generated. In section 4.1, we mention that “we evaluate inferred dependencies on each independent replicate/simulation separately”. Thus, every method is trained on one sequence, like in Tank et al (2018) and Khanna & Tan (2020). This procedure is repeated 5 ($R$) times to evaluate standard deviations of AUROCs, AUPRCs, ACCs, and BAs. That is why we provide 5 sequences for the Lorenz 96 experiment. To make this clearer, we appended a sentence stressing that each method is trained on one sequence to Section 4.1.
> >
> > References:
> >
> > I. Winkler, D. Panknin, D. Bartz, K.-R. Muller, and S. Haufe. Validity of time reversal for testing Granger causality. IEEE Transactions on Signal Processing, 64(11):2746–2760, 2016.
> >
> > S. Haufe, V. V. Nikulin, and G. Nolte. Alleviating the influence of weak data asymmetries on Granger-causal analyses. In Latent Variable Analysis and Signal Separation, pp. 25–33. Springer Berlin Heidelberg, 2012.
> >
> > N. Meinshausen and P. Bühlmann.   Stability selection. Journal of the Royal Statistical Society: Series B (Statistical Methodology), 72(4):417–473, 2010.
> >
> > S. Löwe, D. Madras, R. Zemel, and M. Welling. Amortized causal discovery: Learning to infer causal graphs from time-series data, 2020. arXiv:2006.10833.
> >
> > A. Tank, I. Covert, N. Foti, A. Shojaie, and E. Fox. Neural Granger causality for nonlinear time series, 2018. arXiv:1802.05842.
> >
> > S. Khanna and V. Y. F. Tan. Economy statistical recurrent units for inferring nonlinear Granger causality. In International Conference on Learning Representations, 2020.

---

> > > ### Comment · AnonReviewer2 · 2020-11-24
> > > **About Alg.1**
> > >
> > > Thank you for your response, additional explanation and experiments.
> > > The relationship between the results of prediction error and causality accuracy is interesting (I consider the prediction error should be used for sanity check).
> > >
> > > I understood Alg. 1, but the contribution of stability-based thresholding is still unclear.
> > > 2.2.2. described related work in stability-based selection procedure, but is there any contribution of your work? I recommend changing the title of Alg.1 to be more specific and to present the novelty of your method.

---

> > > > ### Author Response · Authors · 2020-11-24
> > > > **About Alg. 1 Response**
> > > >
> > > > Thank you for your response.
> > > >
> > > > We agree that the prediction error could be used as a proxy for the inferential performance, however, as seen in our experiments and mentioned in the prior work, it is not a perfect proxy.
> > > >
> > > > Regarding the Alg. 1 and its novelty. Stability-based model selection has been used before on a range of tasks, e.g. clustering (Ben-Hur et al, 2002; Lange et al, 2003) and penalised regression model selection (Sun et al, 2013). To the best of our knowledge, none of these works (Ben-Hur et al, 2002; Lange et al, 2003; Meinshausen & Bühlmann, 2010; Sun et al, 2013) focused on sequential data (not to mention GC inference). Implementing the vanilla stability selection e.g. by Meinshausen and Bühlmann (2010) would be prohibitively costly due to the need to retrain many models on bootstrap re-samples of the data for different hyperparameter values. Alg. 1 alleviates this issue by combining the time reversal trick (Winkler et al, 2016) with the general idea of inspecting the model stability. So, this algorithm is particularly tailored towards the concrete inference problem described in the paper.
> > > >
> > > > Thank you for the feedback. We will certainly consider a more specific name for the algorithm.
> > > >
> > > > References:
> > > >
> > > > A. Ben-Hur, A. Elisseeff, and I. Guyon. A stability based method for discovering structure in clustered data. Pacific Symposium on Biocomputing, pp. 6–17, 2002.
> > > >
> > > > T. Lange, M. L. Braun, V. Roth, and J. M. Buhmann. Stability-based model selection. In Advances in Neural Information Processing Systems, pp. 633–642, 2003.
> > > >
> > > > N. Meinshausen and P. Bühlmann. Stability selection. Journal of the Royal Statistical Society: Series B (Statistical Methodology), 72(4):417–473, 2010.
> > > >
> > > > W. Sun, J. Wang, and Y. Fang. Consistent selection of tuning parameters via variable selection stability. Journal of Machine Learning Research, 14(1):3419–3440, 2013.

---

### Author Response · Authors · 2020-11-16
**To All Reviewers: Summary of Revisions**

Dear reviewers,

We would like to thank all of you for the constructive feedback. Below is a summary of revisions addressing some of your concerns:
- We stressed the differences and connections to prior work on time-reversed Granger causality.
- We solidified the presentation of Algorithm 1 to make the text better connected with the pseudo-code, explaining all of the notation that was unclear.
- We added a discussion of the computational complexity of our model compared to the baselines and provided running times of all methods.
- We added the investigation of the prediction error of models on held-out data.
- We performed an ablation experiment on the Lorenz 96 system with varying values of the forcing constant.
- We added a comparison with dynamic Bayesian networks, which are a more conventional approach to structure learning in sequential data.

Please see our point-by-point responses to everyone of you for the detailed discussion.

---

### Decision · Program_Chairs · 2021-01-07
**Final Decision**

**Decision:**

Accept (Poster)

**Comment:**

The proposed approach is interesting and is differentiated enough from the recent body of work on Neural Network Granger causal modeling as it offers a mechanism for detecting signs of causality.

The authors have satisfactorily addressed the points raised in the reviews. In particular relationship with prior work and novelty of the contributions are now clearly articulated. The added discussion on the superiority of TCDF on simulated fMRI experiments is insightful. Though prediction error is only a proxy for the task at hand, the readers will appreciate the added evaluation.

The proposed approach to stability evaluation leveraging the time-reversal trick is novel and particularly pertinent, and could motivate some interesting follow-up work on this topic. It is also important that the authors have characterized the computational advantage of the approach.